# PERSONALIZED FEDERATED COMPOSITE LEARNING WITH FORWARD-BACKWARD ENVELOPES

## ABSTRACT

Federated composite optimization (FCO) is an optimization problem in federated learning whose loss function contains a non-smooth regularizer. It arises naturally in the applications of federated learning (FL) that involve requirements such as sparsity, low rankness, and monotonicity. In this study, we propose a personalization method, called pFedFBE, for FCO by using forward-backward envelope (FBE) as clients' loss functions. With FBE, we not only decouple the personalized model from the global model, but also allow personalized models to be smooth and easily optimized. In spite of the nonsmoothness of FCO, pFedFBE shows the same convergence complexity results as FedAvg for FL with unconstrained smooth objectives. Numerical experiments are shown to demonstrate the effectiveness of our proposed method.

## 1 INTRODUCTION

Federated learning (FL) is originally proposed by (McMahan et al., 2016) to solve learning tasks with decentralized data arising in various applications. For example, data are generated from medical institutions which can not share its data to each other due to confidentiality or legal constraints. Instead of accessing all the data sets, different institutions or clients are under the coordination of a central server and the central server aggregates the local information to train a global model. Similar methodologies have been investigated in the literature of decentralized optimization (Colorni et al., 1991; Boyd et al., 2011; Yang et al., 2019). For more introductions and open problems in the field of federated optimization, we refer to the review articles (Kairouz et al., 2021; Wang et al., 2021).

The local loss functions of FL can be nonsmooth. In particular, it is a summation of a smooth function and a nonsmooth regularizer, where the regularizer is used to promote certain structure of the optimal parameters such as sparisity, low rankness, total variation, and additional constraints on the parameters. This has motivated the recent study of the federated setting of composite optimization (Yuan et al., 2021). The mathematical formulation of FCO is to optimize

$$\min_{w \in \mathbb{R}^d} \quad f(w) := \frac{1}{N} \sum_{i=1}^{N} \left( f_i(w) + h(w) \right), \tag{1}$$

where $f_i(w) = \mathbb{E}_{\xi_i} \tilde{f}_i(w, \xi_i)$ or its empirical version $f_i(w) = \frac{1}{|\mathcal{D}^i|} \sum_{\xi_i \in \mathcal{D}^i} \tilde{f}_i(w, \xi_i)$ is a smooth function with local dataset $\mathcal{D}^i$, and $h : \mathbb{R}^d \to \mathbb{R}$ is a nonsmooth but convex regularizer. Besides, we assume that the proximal operator of $h$, $\text{prox}_h(w) := \arg\min_{u \in \mathbb{R}^d} \quad h(u) + \frac{1}{2}\|u - w\|^2$ has closed-form expressions and is easy to compute. The difference to the centralized setting is that $\mathcal{D}^i$ is local data of client $i$ and the data distribution of each client may not be the same. Optimizing FL with unconstrained smooth objectives, i.e., problem (1) with $h \equiv 0$, has been extensively studied in the literature, e.g., FedAvg (McMahan et al., 2016), FedProx (Li et al., 2020), SCAFFOLD (Karimireddy et al., 2020b) and MIME (Karimireddy et al., 2020a), to name a few. When $h \neq 0$, FedDual (Yuan et al., 2021), FedDR (Tran Dinh et al., 2021) and FedADMM (Wang et al., 2022) are developed. One of the challenges of these algorithms is the heterogeneity of the local dataset $\mathcal{D}^i$, where the distribution of $\mathcal{D}^i$ is none-identical. The model parameter $w$ learned by minimizing $f(w)$ may perform poorly for each client. If each client learns its parameter by its own data, the local model parameters may not generalize well due to the insufficient data. For the case $h \equiv 0$, in order to learn the global parameter and the local parameters jointly, the concept of the personalized FL has been

studied, e.g., (Smith et al., 2017; Hanzely and Richtárik, 2020; Hanzely et al., 2020; Fallah et al., 2020b; Mansour et al., 2020; Chen and Chao, 2021; T Dinh et al., 2020). To our best knowledge, there is no paper which directly investigating the personalization techniques to FCO, although the above personalized methods may be generalized.

**Our Contributions.** In this paper, we consider constructing the personalized model for the FCO (1), while existing methods mentioned above all conduct personalizations to FL with unconstrained smooth objectives, i.e., $h = 0$. Although their approaches may be generalized to (1) by replacing gradients with subgradients, the computational results could be worse due to the slow convergence of subgradient methods. The main contributions are summarized as follows.

- We present a personalized model by utilizing forward-backward envelope (FBE) for FCO (1), called pFedFBE. As a generalized of the Moreau envelope of the Moreau envelope under the Bregman distance (Liu and Pong, 2017), FBE is smooth and has explicit forms of gradients, which is a crucial benefit compared to the Moreau envelope. Analogous to the personalized method by the Moreau envelope (T Dinh et al., 2020), our proposed method is able to obtain both global parameters for generalization and local parameters for personalization. To our best knowledge, this is the first work to investigate personalizations to FCO.

- Based on FBE, the local loss functions and global loss function are smooth and hence FedAvg can be used to solve the resulting model. By applying FedAvg, the optimization process of our proposed personalized model can be regarded as several local variable-metric proximal gradient updates followed by a global aggregated parameter. The variable-metric proximal gradient steps are able to protect the local information, and the aggregation steps guarantee the total loss is minimized at the aggregated parameter. A proper choice of parameter of FBE allows local parameter to move towards their own models and not to go far away from the global parameter. An algorithm, called pFedFBE, is proposed. We show its convergence for the nonconvex $f_i$ under mild assumptions. The complexity result matches the standard results of FedAvg for unconstrained smooth FL.

- Based on the properties of FBE, the convergence rate of pFedFBE match with the standard analysis of applying FedAvg under standard assumptions over $f$, $h$ and the stochasticity. Numerical experiments on various applications are performed to demonstrate the effectiveness of our proposed personalized model.

**Notations.** For a vector $w \in \mathbb{R}^d$ or a matrix $H \in \mathbb{R}^{d \times d}$, we use $\|w\|$ and $\|H\|$ to denote its $\ell_2$ norm and Frobenius norm, respectively. For a smooth function $f : \mathbb{R}^d \to \mathbb{R}$, $\nabla f(x)$ and $\nabla^2 f(x)$ represent its gradient and Hessian at $x$, respectively. For a nonsmooth and convex function $h$, we denote by $\partial h(x)$ its subgradients at $x$. We use $|\mathcal{D}|$ to denote the cardinality of a set $\mathcal{D}$.

## 2 PERSONALIZED FEDERATED LEARNING WITH FORWARD-BACKWARD ENVELOPE (PFEDFBE)

The personalized FedAvg (Per-FedAvg) (Fallah et al., 2020b) and personalized FL with Moreau envelope (pFedMe) (T Dinh et al., 2020) are proposed to deal with the data heterogeneity for the smooth setting, i.e., $h = 0$. For pFedMe, the local model is constructed based on the Moreau envelope of $f_i$, namely,

$$\hat{F}_i(w) = \min_{\theta_i \in \mathbb{R}^d} \ f_i(\theta_i) + \frac{\lambda}{2}\|\theta_i - w\|^2. \tag{2}$$

Then, the resulting personalized model is a bi-level problem:

$$\min_{w \in \mathbb{R}^d} \ \hat{F}(w) := \frac{1}{N} \sum_{i=1}^{N} \hat{F}_i(w). \tag{3}$$

Solving (3) will give both the global parameter $w$ and local personalized parameter $\theta_i(w) := \text{prox}_{f_i/\lambda}(w) := \arg\min_{\theta_i \in \mathbb{R}^d} \ f_i(\theta_i) + \frac{\lambda}{2}\|\theta_i - w\|^2$. A crucial benefit of optimizing with the Moreau envelope $\hat{F}_i$'s lies on the flexible choices of $\lambda$. When $\lambda = \infty$, we have $\hat{F}_i(w) = f_i(w)$ and $\theta_i(w) = w$, which means no personalization is introduced. If $\lambda = 0$, $\hat{F}_i(w)$ is a constant function taking value $f(\theta_i(w))$ with $\theta_i(w) \equiv \arg\min_{\theta_i \in \mathbb{R}^d} f_i(\theta_i)$. In this case, there is only personalization

and no federation. Hence, they claim that a proper $\lambda \in (0, \infty)$ will introduce both federation and personalization.

Since there is no explicit solution of the inner problems, it is proposed to use multiple gradient steps to get an estimation of gradient $\nabla \hat{F}_i(w) = \lambda(w - \text{prox}_{f_i/\lambda}(w))$. For the convergence, they need this estimation satisfying certain accuracy. Once the gradient of $\hat{F}_i(w)$ is available, the existing federated optimization algorithms can be adopted to obtain a global parameter $w$ and locally personalized parameters $\theta_i(w)$.

We note that the Moreau envelope for a nonsmooth function also exists (Rockafellar and Wets, 2009) and pFedMe can be applied to our setting (1). However, obtaining the Moreau envelope needs to solve a nonsmooth problem, which may be costly due to the absence of explicit expressions. Note that the inner problem (3) should be solved to a certain accuracy to guarantee the convergence (T Dinh et al., 2020). The bi-level model (3) will be time-consuming to solve and may not be ideal in this setting.

## 2.1 PROBLEM FORMULATION

Since the Moreau envelope does not obey explicit expressions, we aim to find efficient envelopes not only enjoying simpler formulation, but also sharing similar properties of both federations and personalizations. For composite optimization, a famous generalization of Moreau envelope is called FBE (Stella et al., 2017; Liu and Pong, 2017). Specifically, the FBE of $f_i + h$ is defined as

$$F_i(w) := \min_{\theta_i \in \mathbb{R}^d} \; f_i(w) + \langle \nabla f_i(w), \theta_i - w \rangle + h(\theta_i) + \frac{\lambda}{2} \|\theta_i - w\|^2. \tag{4}$$

When the proximal operator of $h$ has a closed-form solution, the above envelope can be equivalently written as (Stella et al., 2017)

$$F_i(w) = f_i(w) - \frac{1}{2\lambda}\|\nabla f_i(w)\|^2 + H(w - \frac{1}{\lambda}\nabla f_i(w)), \tag{5}$$

where $H(w) := \min_{\theta \in \mathbb{R}^d} \; h(\theta) + \frac{\lambda}{2}\|\theta - w\|^2 = h(\text{prox}_{h/\lambda}(w)) + \frac{\lambda}{2}\|\text{prox}_{h/\lambda}(w) - w\|^2$. Assuming the gradient of $f_i$ is Lipschitz continuous with modulus $L$, i.e., $\|\nabla f_i(\theta) - \nabla f_i(w)\| \leq L\|\theta - w\|$, $\forall \theta, w, i$, the function $F_i(w)$ is continuously differentiable for any $\lambda > L$, with gradients

$$\nabla F_i(w) = \lambda(I - \frac{1}{\lambda}\nabla^2 f_i(w))(w - \text{prox}_{\frac{1}{\lambda}h}(w - \frac{1}{\lambda}\nabla f_i(w))). \tag{6}$$

Compared with the Moreau envelope, the gradient of FBE is of closed form and can be calculated with much less computational costs. Furthermore, when $\lambda > L$, the set of global minimizers of $F_i$ equals to that of $f_i + h$.

With the definition of FBE (4), our personalized model for FCO (1) is

$$\min_{w \in \mathbb{R}^d} \; F(w) := \frac{1}{N}\sum_{i=1}^{N} F_i(w). \tag{7}$$

Similar to pFedMe, one can obtain both global parameter $w$ and local personalized parameter

$$\theta_i(w) := \text{prox}_{\frac{1}{\lambda}h}(w - \nabla f_i(w)) \tag{8}$$

by solving (7). When $\lambda = \infty$, $F_i(w) = f_i(w) + h(w)$, $\theta_i(w) = w$. That is to say, problem (7) reduces to the original problem (1) and there is no personalization. If $\lambda = 0$, then $\theta_i(w) = \arg\min_{\theta_i} \; \nabla f_i(w)^\top(\theta_i - w) + h(\theta_i)$, which is not constant function if $\nabla f_i(w)$ depends on $w$. In the extreme case of linear $f_i$, $\theta_i(w)$ will be a constant function taking value $\arg\min_{\theta_i} \; f_i(\theta_i) + h(\theta_i)$. Then, $\theta_i(w)$ will be the best personalization parameter and no federation introduced. As $\lambda = \infty$ results in only federation, we claim $\lambda \in (0, \infty)$ will allow both federations and personalizations. Other than the linear case, if $f_i$ is a quadratic function with Hessian matrix being $\lambda_0 I$ ($\lambda_0 > 0$), then setting $\lambda = \lambda_0$ will result in the perfect personalization and no federation. In this case, a $\lambda \in (\lambda_0, \infty)$ will guarantee both federations and personalizations. We also note that $\lambda > 0$ is needed for the smoothness. Otherwise, problem (4) is not strongly convex and $F_i$ will not be smooth.

Besides, compared with original model (1), i.e., $\lambda = \infty$, the objective function of our new model (7) is smooth and hence easier to optimize. Basically, our new model takes the solution of (7) as an initial point and slightly update it with respect to their own data by performing one proximal gradient step. The benefits compared with the original Moreau envelope lie on the explicit expressions of $\theta_i(w)$ and $\nabla F_i(w)$, given in (8) and (6), respectively.

To summarize, our proposed model (7) has the following advantages:

- The flexible choice of $\lambda$ allows the user-defined trade-off between FL and personalization.
- $F_i$ is smooth while shares the same optimizers with the nonsmooth function $f_i + h$.
- Although the Moreau envelope of $f_i + h$ is smooth, the calculations of gradients are expensive compared with FBE $F_i$. Both the gradients $\nabla F_i(w)$ and local personalized parameters $\theta_i(w)$ has explicit expressions whenever $\mathrm{prox}_h$ has simple and closed-form solution.

## 2.2 PFEDFBE: ALGORITHM

In this subsection, we conduct an algorithm, called pFedFBE, to solve our proposed model (7). Since $F_i(w)$ is smooth and its gradient is with explicit expression, solving (7) falls into the classic FL setting. One may utilize the existing methods (McMahan et al., 2016; Li et al., 2020; Karimireddy et al., 2020b;a).

We now describe how to use FedAvg to solve our proposed model (7). At $k$-th step, the server randomly select a subset of clients, denoted by $\mathcal{S}_k$. Each selected client is initialized with $w_k$ and optimized by $R$ local updates. By collecting all local parameters $\{w_{k,R}^i\}_{i \in \mathcal{S}_k}$, the server updates its model by $w_{k+1} = \frac{1}{|\mathcal{S}_k|} \sum_{i \in \mathcal{S}_k} w_{k,R}^i$. Let us introduce the details of local updates. Since the full batch gradient is costly, we take a minibatch $\mathcal{D}_k^i \subset \mathcal{D}^i$ and calculate the unbiased minibatch gradient

$$\nabla f_i(w_{k,t}^i) \approx \nabla \tilde{f}_i(w_{k,t}^i, \mathcal{D}_{k,t}^i) := \frac{1}{|\mathcal{D}_{k,t}^i|} \sum_{\xi_i \in \mathcal{D}_{k,t}^i} \nabla \tilde{f}_i(w_{k,t}^i, \xi_i).$$

From the expression of $\nabla F_i(w)$, we need to compute the Hessian of $f_i$ as well. By using another minibatch $\hat{\mathcal{D}}_{k,t}^i \subset \mathcal{D}^i$, the unbiased estimated Hessian is

$$\nabla^2 f_i(w_{k,t}^i) \approx \nabla^2 \tilde{f}_i(w_{k,t}^i, \hat{\mathcal{D}}_k^i) := \frac{1}{|\hat{\mathcal{D}}_{k,t}^i|} \sum_{\xi_i \in \hat{\mathcal{D}}_k^i} \nabla^2 \tilde{f}_i(w_{k,t}^i, \xi_i).$$

Plugging the estimations into (6), we obtain the estimated gradient

$$\nabla F_i(w_{k,t}^i) \approx g_i(w_{k,t}^i) := \lambda(I - \frac{1}{\lambda}\nabla^2 \tilde{f}_i(w_{k,t}^i, \hat{\mathcal{D}}_{k,t}^i))(w_{k,t}^i - \mathrm{prox}_{\frac{1}{\lambda}h}(w_{k,t}^i - \frac{1}{\lambda}\nabla \tilde{f}_i(w_{k,t}^i, \mathcal{D}_{k,t}^i))).$$
$$(9)$$

Note that $g_i(w_{k,t}^i)$ is biased due to the nonlinearity of $\mathrm{prox}_{\frac{1}{\lambda}h}$. After obtaining the estimated gradient of $F_i$, we do $R$-step stochastic gradient descent with a fixed step size $\eta > 0$, namely,

$$w_{k,0}^i := w_k, \ w_{k,t+1}^i = w_{k,t}^i - \eta g_i(w_{k,t}^i), \ i = 0, \dots, R-1.$$

The detailed algorithm is presented in Algorithm 1. Note that the computations of Hessian may be costly. The approximations to the Hessian are developed in (Finn et al., 2017; Nichol et al., 2018; Fallah et al., 2020a). To make the computations afordable, we use the following approximations in our numerical experiments:

$$\nabla^2 f(w)[u] \approx (\nabla f(w + tu) - \nabla f(w))/t,$$

where $t$ is a small positive number. Hence, two mini-batch gradient evaluations of $\tilde{f}_i$ are needed for estimating $\nabla F_i$. Similar to (Fallah et al., 2020b, Section 5), the proximal gradient $(w_{k,t}^i - \mathrm{prox}_{\frac{1}{\lambda}h}(w_{k,t}^i - \frac{1}{\lambda}\nabla \tilde{f}_i(w_{k,t}^i, \mathcal{D}_{k,t}^i)))$ can also serve as an efficient estimate of $\nabla F_i(w_{k,t}^i)$ when $\lambda > L$.

## 3 CONVERGENCE

In this section, we present the convergence results of the proposed pFedFBE, i.e., Algorithm 1. Let us start with the following necessary assumptions.

---
**Algorithm 1:** pFedFBE for solving (7)

---
**input:** Initial point $w_0$, personalization parameter $\lambda$ and learning rate $\eta$.
**for** $k = 0, 1, \ldots, K - 1$ **do**
    Sample a subset $\mathcal{S}_k$ of clients
    **for** *client* $i \in \mathcal{S}_k$ *in parallel* **do**
        Initialize local model $w_{k,0}^i = w_k$
        **for** $t = 0, 1, \ldots, R - 1$ **do**
            Sample two minibatches $\mathcal{D}_{k,t}^i$ and $\hat{\mathcal{D}}_{k,t}^i$ from $\mathcal{D}^i$
            Calculate the personalized parameter $\theta_i(w_{k,t}^i) = \text{prox}_{\frac{1}{\lambda} h}(w_{k,t}^i - \frac{1}{\lambda} \nabla \tilde{f}_i(w_{k,t}^i, \mathcal{D}_{k,t}^i))$
            Compute local stochastic gradient $g_i(w_{k,t}^i)$ by following (9)
            Perform local update $w_{k,t+1}^i = w_{k,t}^i - \eta g_i(w_{k,t}^i)$

Aggregate local parameters and set $w_{k+1} = \frac{1}{|\mathcal{S}_k|} \sum_{i \in \mathcal{S}_k} w_{k,R}^i$

---

**Assumption 1** *Let $f_i = 1, 2, \ldots, N$ and $h$ be the local functions in FCO* (1).

(A1) *Functions $f_i, i = 1, 2, \ldots, N$ are smooth, $h$ is proper, closed, convex with cheap proximity operator, and functions $f_i + h, i = 1, 2, \ldots, N$ are coercive (i.e., $\liminf_{\|w\| \to \infty} \frac{f_i(x) + h(x)}{\|x\|} > 0$).*

(A2) *The functions $f_i, \; i = 1, 2, \ldots, N$ are twice continuously differentiable and $L$-smooth, and the gradients $\nabla f_i(w)$ and subgrdients $\tilde{\partial} h(w) \in \partial h(w)$ are bounded by a positive constant $B$ in a compact ball $\mathcal{C}$ containing $\{w_{k,t}^i\}_{k,t,i}$ generated by Algorithm 1, namely, for all $i$,*

$$\|\nabla f_i(w) - \nabla f_i(u)\| \leq L \|w - u\|, \; \forall w, u \in \mathbb{R}^d,$$
$$\|\nabla f_i(w)\| \leq B, \; \forall w \in \mathcal{C},$$
$$\left\| \tilde{\partial} h_i(w) \right\| \leq B, \; \forall w \in \mathcal{C}, \; \tilde{\partial} h_i(w) \in \partial h_i(w).$$

(A3) *For each $i \in \{1, \ldots, N\}$, the Hessian of the function $f_i$ is $\rho$-Lipschitz continuous, i.e.,*

$$\left\| \nabla^2 f_i(w) - \nabla^2 f_i(u) \right\| \leq \rho \|w - u\|, \; \forall w, u \in \mathbb{R}^d, i.$$

(A4) *For any $w \in \mathbb{R}^d$, the stochastic gradient $\nabla \tilde{f}(w, \xi_i)$ and Hessian $\nabla^2 \tilde{f}_i(w, \xi_i)$, computed with respect to a single data point $\xi_i \in \mathcal{D}_i$, have bounded variance, i.e., for all $i$ and $w$,*

$$\mathbb{E}_{\xi_i} \left[ \left\| \nabla \tilde{f}_i(w, \xi_i) - \nabla f_i(w) \right\|^2 \right] \leq \sigma_G^2,$$
$$\mathbb{E}_{\xi_i} \left[ \left\| \nabla^2 \tilde{f}_i(w, \xi_i) - \nabla^2 f_i(w) \right\|^2 \right] \leq \sigma_H^2.$$

(A5) *For any $w \in \mathbb{R}^d$, the gradient and Hessian of local functions $f_i(w)$ and the global function $f(w) := \sum_{i=1}^N f_i(w)$ satisfy the following conditions*

$$\|\nabla f_i(w) - \nabla f(w)\|^2 \leq \gamma_G^2, \; \left\| \nabla^2 f_i(w) - \nabla^2 f(w) \right\|^2 \leq \gamma_H^2, \; \forall i, w.$$

The smoothness of $f_i$ and cheap proximity operator property of $h$ are standard assumptions in FCO (Yuan et al., 2021). The assumptions (A2) and (A3) hold for sufficiently smooth $f$, which are satisfied by many problems arising from machine learning, such as the federated Lasso problem and the federated matrix completion problem in Section 4. These two assumptions are used for the Lipschitz continuous property of $F_i$. The bounded variance condition (A4) and bounded diversity condition (A5) are crucial to control the bias introduced by the randomness and the heterogeneity of the local clients. Conditions on $f_i$ in Assumption 1 are also made in (Fallah et al., 2020b), and the conditions on $h$ is standard in decentralized composite optimization, see (Zeng and Yin, 2018). Compared with the analysis of pFedMe (T Dinh et al., 2020), we do not need an additional assumption on the exactness of solving the Moreau envelope since the closed-form solutions is avaliable for the FBE.

Based on Assumption 1, FBE $F_i$ has the following desired properties.

**Proposition 1** *(Liu and Pong, 2017, Theorem 3.1) If Assumption* (A1) *holds, then $F_i$ is smooth and level-bounded for all $\lambda > L$. Here, we say a function $\varphi$ is level-bounded if $\{w \in \mathbb{R}^d : \varphi(w) \leq \gamma\}$ is bounded for all $\gamma \in \mathbb{R}$.*

The level boundedness and the smoothness ensure the existence of the minimizer of $F$. The next lemma establishes the gradient Lipschitz continuous property of $F_i$ and $F$.

**Lemma 1** *Suppose that Assumptions (A1)-(A2) hold. If $\lambda > L$, the gradient $\nabla F_i$ is Lipschitz continuous with modulus $L_F := \frac{2\rho B + (\lambda + L)(2\lambda + L)}{\lambda}$ over the set $\mathcal{C}$.*

The assumption (A5) gives the bounded divergence between $f_i$ and $f$. We show that the divergence of the local gradient of FBE is also bounded.

**Lemma 2** *Suppose that the assumptions (A2) and (A5) hold. Then, for $\lambda > L$, we have*

$$\frac{1}{N} \sum_{i=1}^{N} \|\nabla F_i(w) - \nabla F(w)\|^2 \leq \gamma_F := 24\gamma_G^2 + \frac{12B^2}{\lambda^2}\gamma_H^2.$$

With the variance assumptions (A4) of $f_i$, we show the estimations on $F_i$ in the following lemma.

**Lemma 3** *Let $\mathcal{D}_1, \mathcal{D}_2 \subset \mathcal{D}$ be the sample set and independent to each other. Suppose that (A4) holds. Then for any $\lambda > L$, it holds*

$$\|\mathbb{E}\left[g_i(w, \mathcal{D}_1, \mathcal{D}_2) - \nabla F_i(w)\right]\| \leq \frac{2}{\sqrt{|\mathcal{D}_1|}}\sqrt{1 - \frac{|\mathcal{D}_1| - 1}{|\mathcal{D}| - 1}}\sigma_G,$$

$$\mathbb{E}\left[\|g_i(w, \mathcal{D}_1, \mathcal{D}_2) - \nabla F_i(w)\|^2\right] \leq \sigma_F^2 := \frac{12}{|\mathcal{D}_1|}\sigma_G^2 + \frac{12B^2}{|\mathcal{D}_2|\lambda^2}\sigma_H^2 + \frac{1}{|\mathcal{D}_1||\mathcal{D}_2|\lambda^2}\sigma_G^2\sigma_H^2.$$

The above lemma tells us the error of the stochastic gradient of $F_i$ can be controlled by the variances of the stochastic gradient and Hessian of $f_i$. Note that the estimated gradient of $F_i$ is biased unless using the full batch $\mathcal{D}$.

**Lemma 4** *Let $\{w_{k,t}^i\}$ be the iterates generated by Algorithm 1. Suppose that Assumption 1 holds. Assume that $\lambda > L$ and $\eta < \frac{1}{10L_F R}$, we have*

$$\mathbb{E}\left[\frac{1}{N}\sum_{i=1}^{N}\left\|w_{k,t}^i - w_{k,t}\right\|\right] \leq 4\eta t\left(\sigma_F + \gamma_F\right), \tag{10}$$

$$\mathbb{E}\left[\frac{1}{N}\sum_{i=1}^{N}\left\|w_{k,t}^i - w_{k,t}\right\|^2\right] \leq 48tR\eta^2(\gamma_F^2 + 4\sigma_F^2), \tag{11}$$

*where $w_{k,t} := \frac{1}{N}\sum_{i=1}^{N}w_{k,t}^i$.*

The above lemma presents the consensus error induced by the local stochastic gradient updates, which is proportional to the local step size $\eta$. With the above preparations, we show the following convergence of Algorithm 1.

**Theorem 1** *Consider the objective function $F_i$ defined in (4) for the case that $\lambda > L$. Suppose that Assumption 1 is satisfied, and recall the definitions of $L_F$, $\gamma_F$ and $\sigma_F$, from Lemmas 1, 2 and 3, respectively. Consider running Algorithm 1 for $K$ rounds with $R$ local updates in each round and with $\eta \leq \frac{1}{10RL_F}$. Then, the following first-order stationary condition holds*

$$\frac{1}{RK}\sum_{k=0}^{K-1}\sum_{t=0}^{R-1}\mathbb{E}\left[\|\nabla F\left(\bar{w}_{k+1,t}\right)\|^2\right] \leq \frac{4\left(F\left(w_0\right) - F^*\right)}{\eta RK} + 1600\eta^2 R^2(\gamma_F^2 + 4\sigma_F^2)$$

$$+ 8L_F\eta\left(\frac{N-S}{S(N-1)}\gamma_F^2 + \sigma_F^2\right) + 16r\sigma_G^2,$$

*where $\bar{w}_{k+1,t} = \frac{1}{S}\sum_{i\in\mathcal{S}_k}w_{k+1,t}^i$ with $\bar{w}_{k+1,0} = w_k$ and $\bar{w}_{k+1,R} = w_{k+1}$, and $r = \max_{k,t,i}\left(1 - \frac{|\mathcal{D}_{k,t}^i|-1}{|\mathcal{D}^i|-1}\right)/|\mathcal{D}_{k,t}^i|$.*

**Remark.** Theorem 1 presents the results of using fixed step size. One can easily extend to the setting of diminishing step size. Due to the bias of the estimated gradient of $F_i$, the squared norm of $F$ at $\bar{w}_{k,t}$ will converge to a ball of center $0$ and radius $16r\sigma_G^2$. When full-batch gradients are used, this radius diminishes. By taking $\eta = 1/\sqrt{RK}$, the convergence speed of the squared norm of expected gradient is $\mathcal{O}(\frac{1}{\sqrt{RK}})$, which is similar to (Deng et al., 2020; Reddi et al., 2021). Since $F_i$ is smooth, the advanced algorithms, such as FedProx (Li et al., 2020) and SCAFFOLD (Karimireddy et al., 2020b) can also be adopted for better complexity results.

## 4 NUMERICAL EXPERIMENTS

### 4.1 FEDERATED LASSO

Federated Lasso was considered in (Yuan et al., 2021), which is to recover the sparse ground-truth signal from observations. The mathematical formulation is

$$\min_{w \in \mathbb{R}^d,\, b \in \mathbb{R}} \frac{1}{N} \sum_{j=1}^{N} \left( \sum_{i=1}^{n_j} ((x_j^{(i)})^\top w + b - y_j^{(i)})_2^2 \right) + \lambda \|w\|_1,$$

where $N$ is the number of clients and client $j$ has $n_j$ observation pairs $\{(x_j^{(i)}, y_j^{(i)})\}_{i=1}^{n_j}$. We set $\lambda = 0.1$ in our numerical experiments.

**Synthetic Dataset Descriptions.** We consider both i.i.d. setting and non-i.i.d. setting.

(I) We generate the ground truth $w_{\text{real}} \in \mathbb{R}^{1024}$ with $d_1 = 992$ ones and $d_0 = 32$ zeros, namely,

$$w_{\text{real}} = [\mathbf{1}_{d_1}^\top, \mathbf{0}_{d_0}^\top]^\top,$$

and ground truth $b_{\text{real}} = 0$. The observations $(x, y)$ are generated as follows to simulate the heterogeneity among clients. Let $\left(x_j^{(i)}, y_j^{(i)}\right)$ denotes the $i$-th observation of the $j$-th client. For each client $j$, we first generate and fix the mean $\mu_j \sim \mathcal{N}\left(0, I_{d \times d}\right)$. Then, we sample $n_j$ pairs of observations following

$$x_j^{(i)} = \mu_j + \delta_j^{(i)}, \quad \text{where } \delta_j^{(i)} \sim \mathcal{N}\left(\mathbf{0}_d, I_{d \times d}\right) \text{ are i.i.d., for } i = 1, \ldots, n_j$$

$y_j^{(i)} = w_{\text{real}}^\top x_j^{(i)} + \varepsilon_j^{(i)}$, where $\varepsilon_j^{(i)} \sim \mathcal{N}(0, 1)$ are i.i.d., for $i = 1, \ldots, n_j$. We generate $N = 30$ training clients where each client possesses 128 pairs of samples. There are 3840 training samples in total.

(II) The ground truth $w_i \in \mathbb{R}^{1024}$ of each client are constructed as follows:

$$w_{\text{real}} = [\mathbf{1}_{d_1}^\top, 0, \ldots, 0, 0.5 \times 1_{i_1}, 0, \ldots, 0, 0.5 \times 1_{i_{d_0}}, 0, \ldots, 0]^\top,$$

where $i_1, \ldots i_{d_0}$ are uniformly drawn from the $\{d_1 + 1, d_1 + 2, \ldots, 1024\}$ for each client. We set $d_1 = 8$ and $d_0 = 2$. Using the same data generalization process as in (I), 30 training clients where each client possesses 128 pairs of samples are constructed.

The numerical results for settings (I) and (II) are presented in Figures 1, 2, 3, and 4. The precision, recall, density, and F1 indexes are calculated by measuring the difference between the ground truth $w_{\text{real}}$ and the obtained parameters by the algorithms (where any element with absolute value less than $0.01$ is regraded as $0$). We compare with the baseline algorithms, FedAvg (McMahan et al., 2016), Fedmirror (Yuan et al., 2021), FedDual (Yuan et al., 2021), pFedMe (T Dinh et al., 2020), and pFedditto (Li et al., 2021). For all algorithms, we set the maximum number of rounds to 200. In each round, we sample 10 clients and run 20 local iterations with batch size $50$. For all algorithms, we tune to get the best clients' learning rate and keep the remaining parameters as their default values. For both settings (I) and (II), we set the learning rate $\eta = 0.0005$ and $\lambda = 2000$ for pFedFBE. From Figure 1 and 2, we see that our proposed pFedFBE and Fedmirror give the best performances among all algorithms for setting (I). The poor performance of FedDual compared with Fedmirror may be from the multiple local iterations while the number of the local iteration is set to $1$ in (Yuan et al., 2021). For the non-i.i.d. setting (II), Figures 3 and 4 show the results with respect to the personalized

parameters. For those algorithms without personalization, we directly set the personalized parameter as the global parameters. From the test precision and recall, our proposed pFedFBE is able to find all nonzero elements of $w_{\text{real}}$ and do not introduce extra nonzero elements. Although the personalized algorithm pFedditto can recognize all nonzero elements, the zero entries are mistaken as nonzero as well. pFedFBE gives the best F1 score, the test accuracy, and the train loss.

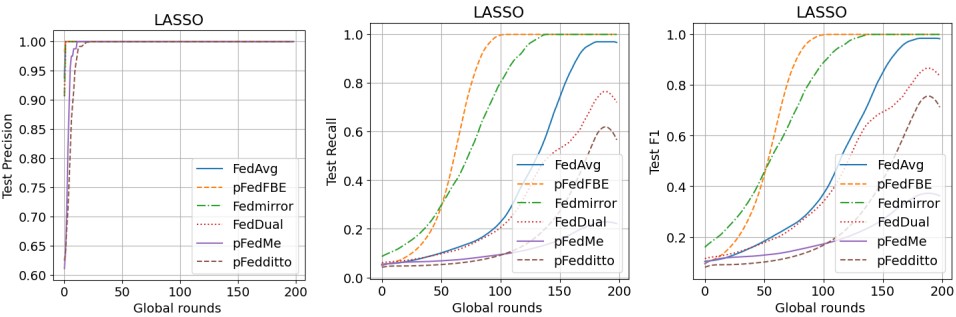

Figure 1: Results for federated Lasso with setting (I).

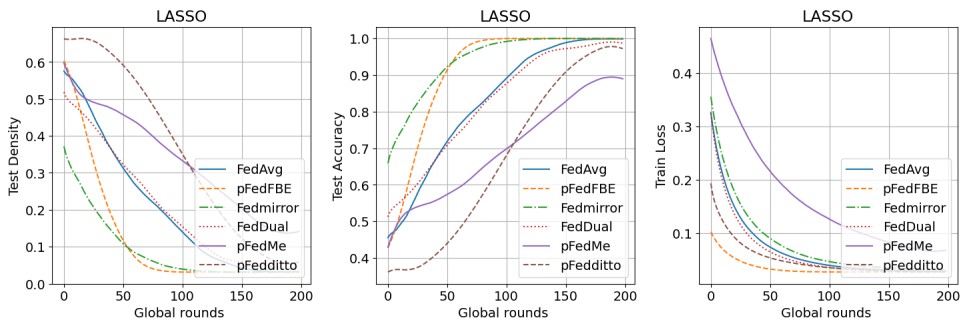

Figure 2: Results (cont.) for federated Lasso with setting (I).

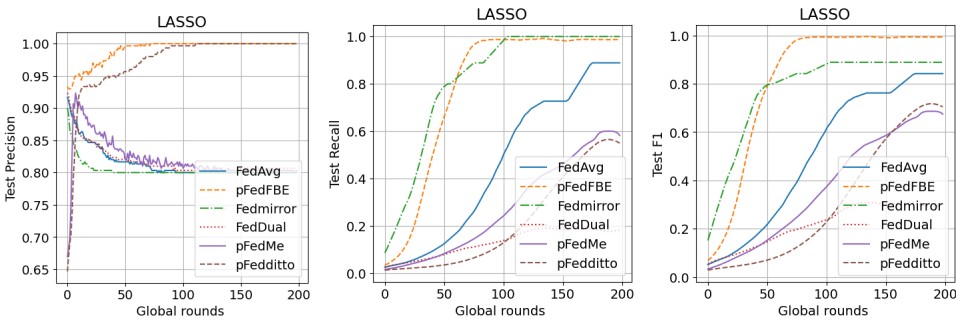

Figure 3: Results for federated Lasso with setting (II).

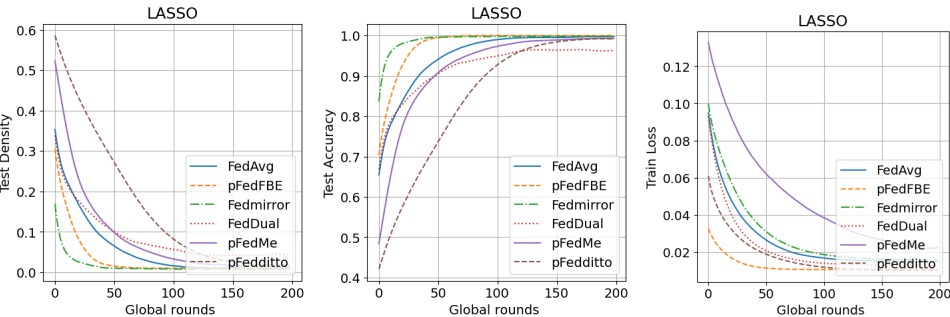

Figure 4: Results (cont.) for federated Lasso with setting (II).

### 4.2 FEDERATED MATRIX COMPLETION

Federated matrix completion (Yuan et al., 2021) can be mathematically formulated as

$$\min_{W\in\mathbb{R}^{d\times d},\, b\in\mathbb{R}} \frac{1}{N}\sum_{j=1}^{N}\left(\sum_{i=1}^{n_j}(\left\langle X_j^{(i)},W\right\rangle + b - y_i)_2^2\right) + \lambda\|W\|_{\mathrm{nuc}},$$

where $N$ is the number of clients, client $j$ has $n_j$ observation pairs $\{(X_j^{(i)},y_j^{(i)})\}_{i=1}^{n_j}$, $\|W\|_{\mathrm{nuc}}$ is the nuclear norm of $W$, and $\lambda > 0$ is a parameter to control the rank of $W$. We set $\lambda = 0.1$ in our numerical experiments.

We only present the numerical results for a non-i.i.d. setting to exhibit the importance of personalization. The setting is as follows. We set the number of clients to 30 and generate a vector $w_j \in \mathbb{R}^{32}$ of the form

$$w_j = [\mathbf{1}_4^\top, 0, \ldots, 0, 0.25 \times 1_{d_0}, 0, \ldots, 0]^\top,$$

where $d_0$ is uniformly drawn from $\{5, 6, \ldots, 32\}$ for each client. After getting $w_j$, we set the diagonal matrix $W_j = \mathrm{diag}(w_j)$ with diagonal elements $w_j$ as the local ground truth. For each client $j$, we first generate and fix the mean $\mu_j \sim \mathcal{N}(0, I_{d\times d})$. Then we sample $n_j$ pairs of observations following

$$x_j^{(i)} = \mu_j + \delta_j^{(i)}, \quad \text{where } \delta_j^{(i)} \sim \mathcal{N}(\mathbf{0}_d, I_{d\times d}) \text{ are i.i.d., for } i = 1, \ldots, n_j$$

$y_j^{(i)} = \left\langle W_j, X_j^{(i)}\right\rangle + \varepsilon_j^{(i)}$, where $\varepsilon_j^{(i)} \sim \mathcal{N}(0,1)$ are i.i.d., for $i = 1, \ldots, n_j$. 128 pairs of samples are generated for each client.

The numerical results are presented in Figure 5. Since FedDual is not comparable to Fedmirror in the case of multiple local iterations, we omit it and add comparisons with pFedprox (Li et al., 2020). Analogous to the Federated Lasso, the total number of global rounds is set to 200. In each round, we randomly select 10 clients and perform 20 local iterations with batch size 50. We only tune to get the best step sizes for each algorithm. The learning rate $\eta$ and the parameter $\lambda$ used for pFedFBE are 0.0005 and 2000, respectively. From Figure 5, the personalized parameters of pFedFBE are able to recover the ground truth rank 5, while all other algorithms fail. Moreover, pFedFBE converges fastest to better training loss, training MSE, and recovery error (which is defined as the Euclidean distance between the local ground truth and the obtained personalized parameters).

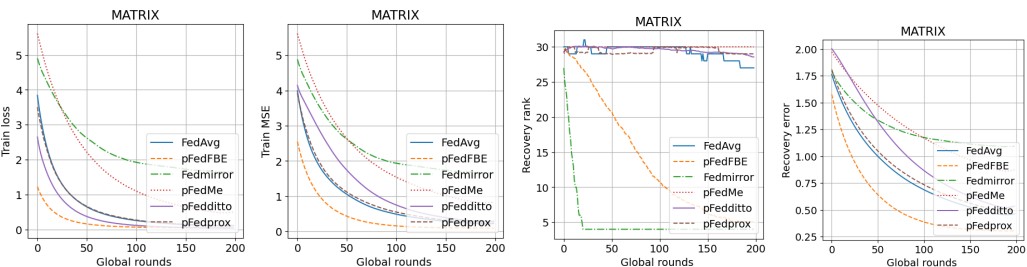

Figure 5: Results for federated matrix completion.

### 4.3 NEURAL NETWORK WITH NONSMOOTH REGULARIZATION

Consider a two-layer deep neural network with hidden layer of size 100, the ReLU activation, and a softmax layer at the end. The numerical results are performed on the Mnist dataset, which consists of 7000 handwritten digit images from 10 classes. We distribute the complete dataset to $N = 20$ clients. To model a heterogeneous setting in terms of local data sizes and classes, each client is allocated a different local data size in the range of $[1165, 3834]$ and only has 2 of the 10 labels. A similar setting is used in (T Dinh et al., 2020). The loss function is a sum of the cross-entropy loss and the nonsmooth $\ell_2$ norm function on the weights. Therefore, the resulting problem takes the composite form (1).

For the numerical tests, we set the total number of global rounds to 100. In each round, we randomly select 10 clients and perform 10 local iterations with batch size 20. For all algorithms, we tune to get the best clients' learning rate and keep the remaining parameters as their default values. We use the learning rate $\eta = 0.005$ and $\lambda = 200$ for pFedFBE. The results are presented in Figure 6, where the

global accuracies are based on the global parameters, and personalized accuracies are computed from the local parameters. We see the algorithms taking the composite structures, pFedFBE, Fedmirror, and FedDual, converge faster than the other algorithms in terms of the global accuracies. Moreover, pFedFBE outperforms Fedmirror and FedDual. For the personalized accuracies, our pFedFBE performs the best although the personalized algorithms could achieve better accuracies.

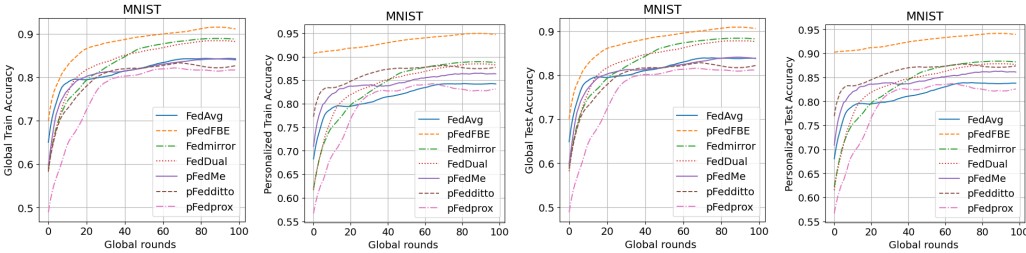

Figure 6: Results for non-iid Mnist dataset

## 5 CONCLUSION

In this paper, we propose a novel personalized method for FCO. The idea is to use the forward-backward envelope arising from the composite optimization. We then utilize the FedAvg algorithm to solve the resulted Federated learning problem with smooth objective functions. Convergence results of the proposed algorithm are shown under standrad assumptions. Numerical experiments on federated lasso, federated matrix completion, and nonsmooth deep neural network outperforms the existing methods.

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

# A APPENDIX

## A.1 PROOF OF LEMMA 1

Note that $\nabla F_i(w) = \lambda(I - \frac{1}{\lambda}\nabla^2 f_i(w))(w - \text{prox}_{\frac{1}{\lambda}h}(w - \frac{1}{\lambda}\nabla f_i(w)))$. It holds for $w_1, w_2 \in \mathcal{C}$ that

$$
\begin{aligned}
&\|\nabla F_i(w_1) - \nabla F_i(w_2)\| \\
=&\lambda \left\| (I - \frac{1}{\lambda}\nabla^2 f_i(w_1))(w_1 - \text{prox}_{\frac{1}{\lambda}h}(w_1 - \frac{1}{\lambda}\nabla f_i(w_1))) \right. \\
&\left. - (I - \frac{1}{\lambda}\nabla^2 f_i(w_2))(w - \text{prox}_{\frac{1}{\lambda}h}(w_2 - \frac{1}{\lambda}\nabla f_i(w_2))) \right\| \\
=&\lambda \left\| \left((I - \frac{1}{\lambda}\nabla^2 f_i(w_1)) - (I - \frac{1}{\lambda}\nabla^2 f_i(w_2))\right)(w_1 - \text{prox}_{\frac{1}{\lambda}h}(w_1 - \frac{1}{\lambda}\nabla f_i(w_1))) \right. \\
&\left. + (I - \frac{1}{\lambda}\nabla^2 f_i(w_2))\left(w_1 - \text{prox}_{\frac{1}{\lambda}h}(w_1 - \frac{1}{\lambda}\nabla f_i(w_1)) - w_2 + \text{prox}_{\frac{1}{\lambda}h}(w_2 - \frac{1}{\lambda}\nabla f_i(w_2))\right) \right\| \\
\leq&\|\nabla^2 f_i(w_1) - \nabla^2 f_i(w_2)\|\|w_1 - \text{prox}_{\frac{1}{\lambda}h}(w_1 - \frac{1}{\lambda}\nabla f_i(w_1))\| \\
&+ \lambda\|I - \frac{1}{\lambda}\nabla^2 f_i(w_2)\| \left( \|w_1 - w_2\| + \|w_1 - \frac{1}{\lambda}\nabla f_i(w_1) - w_2 + \frac{1}{\lambda}\nabla f_i(w_2)\| \right) \\
\leq&\rho\|w_1 - w_2\|\|w_1 - \text{prox}_{\frac{1}{\lambda}h}(w_1 - \frac{1}{\lambda}\nabla f_i(w_1))\| + \lambda\left(1 + \frac{L}{\lambda}\right)\left(2 + \frac{L}{\lambda}\right)\|w_1 - w_2\| \\
\leq&\left(\rho\left(\frac{1}{\lambda}\|\nabla f_i(w_1)\| + \frac{1}{\lambda}\max_{\theta \in \partial h(\text{prox}_{\frac{1}{\lambda}h}(w_1 - \frac{1}{\lambda}\nabla f_i(w_1)))}\|\theta\|\right) + \frac{(\lambda + L)(2\lambda + L)}{\lambda}\right)\|w_1 - w_2\| \\
\leq&\frac{2\rho B + (\lambda + L)(2\lambda + L)}{\lambda}\|w_1 - w_2\|,
\end{aligned}
$$

where the first inequality is due to triangle inequality, the second inequality is from (A3) and the nonexpansive property of $\text{prox}_{\frac{1}{\lambda}h}$, the third inequality is from $w - \text{prox}_{\frac{1}{\lambda}h}(u) \in \partial h(\text{prox}_{\frac{1}{\lambda}h}(u))$, and the last inequality is due to (A2).

## A.2 PROOF OF LEMMA 2

It follows from $\nabla F_i(w) = \lambda(I - \frac{1}{\lambda}\nabla^2 f_i(w))(w - \text{prox}_{\frac{1}{\lambda}h}(w - \frac{1}{\lambda}\nabla f_i(w)))$ that

$$
\nabla F_i(w) - \nabla F(w) = \lambda\left(I - \frac{1}{\lambda}\nabla^2 f(w)\right)r_i + \lambda E_i(w - \text{prox}_{\frac{1}{\lambda}h}(w - \frac{1}{\lambda}\nabla f(w))) + \lambda E_i r_i,
$$

where $E_i = \frac{1}{\lambda}(\nabla^2 f(w) - \nabla^2 f_i(w))$ and $r_i = \text{prox}_{\frac{1}{\lambda}h}(w - \frac{1}{\lambda}\nabla f(w)) - \text{prox}_{\frac{1}{\lambda}h}(w - \frac{1}{\lambda}\nabla f_i(w))$. Due to (A5), it holds

$$
\|E_i\|^2 \leq \frac{1}{\lambda^2}\gamma_H^2. \tag{12}
$$

Using the nonexpansive property of $\text{prox}_{\frac{1}{\lambda}h}$, we have

$$
\|r_i\|^2 \leq \frac{1}{\lambda^2}\|\nabla f_i(w) - f(w)\|^2 \leq \frac{1}{\lambda^2}\gamma_G^2. \tag{13}
$$

Combining (12) and (13), it holds

$$\frac{1}{N}\sum_{i=1}^{N}\|\nabla F_i(w) - \nabla F(w)\|^2$$

$$\leq 3\lambda^2\left((1+\frac{L}{\lambda})^2\frac{1}{N}\sum_{i=1}^{N}\|r_i\|^2 + \frac{4B^2}{\lambda^2}\frac{1}{N}\sum_{i=1}^{N}\|E_i\|^2 + \frac{1}{N}\sum_{i=1}^{N}\|E_i\|^2\|r_i\|^2.\right)$$

$$\leq 3\left((\lambda+L)^2\frac{1}{\lambda^2}\gamma_G^2 + 4B^2\frac{1}{\lambda^2}\gamma_H^2 + \lambda^2\max_i\|E_i\|^2\frac{1}{\lambda^2}\gamma_G^2\right)$$

$$\leq 3\left(4\gamma_G^2 + \frac{4B^2}{\lambda^2}\gamma_H^2 + 4\gamma_G^2\right)$$

$$= 24\gamma_G^2 + \frac{12B^2}{\lambda^2}\gamma_H^2,$$

where the first inequality is from $\|I - \frac{1}{\nabla^2}f(w)\| \leq 1 + \frac{L}{\lambda}$ and $\|w - \text{prox}_{\frac{1}{\lambda}h}(w - \frac{1}{\lambda}\nabla f(w))\| \leq \frac{2B}{\lambda}$, the second inequality is due to (12) and (13), and the last inequality is based on $\|E_i\| \leq \frac{2L}{\lambda}$ and $\lambda > L$.

## A.3 PROOF OF LEMMA 3

From the definition of $g_i(w)$ in (9) with the sample set $\mathcal{D}_1$ for the gradient and the sample set $\mathcal{D}_2$ for the Hessian, we have

$$g_i(w) - \nabla F_i(w) = \lambda e_1(w - \text{prox}_{\frac{1}{\lambda}h}(w - \frac{1}{\lambda}\nabla f_i(w))) + \lambda(I - \frac{1}{\lambda}\nabla^2 f_i(w))e_2 + \lambda e_1 e_2,$$

where $e_1 = \frac{1}{\lambda}(\nabla^2 f_i(w) - \nabla^2\tilde{f}_i(w,\mathcal{D}_2))$ and $e_2 = \text{prox}_{\frac{1}{\lambda}h}(w - \frac{1}{\lambda}\nabla f_i(w)) - \text{prox}_{\frac{1}{\lambda}h}(w - \frac{1}{\lambda}\nabla\tilde{f}_i(w,\mathcal{D}_1))$. Let us estimate $e_1$ and $e_2$ first. Due to (A4), we have

$$\mathbb{E}[e_1] = 0, \quad \mathbb{E}[\|e_1\|^2] \leq \frac{1}{|\mathcal{D}_2|\lambda^2}\sigma_H^2, \tag{14}$$

where $|\mathcal{D}_2|$ is the number of samples. For $e_2$, it follows from the nonexpansive property of $\text{prox}_{\frac{1}{\lambda}h}$ that

$$\|\mathbb{E}[e_2]\| \leq \mathbb{E}\left[\frac{1}{\lambda}\|\nabla f_i(w) - \nabla\tilde{f}_i(w,\mathcal{D}_1)\|\right] \leq \frac{1}{\lambda\sqrt{|\mathcal{D}_1|}}\sqrt{1 - \frac{|\mathcal{D}_1|-1}{|\mathcal{D}|-1}}\sigma_G, \tag{15}$$

where the second inequality is from (A4) and the variance of sampling without replacement. Similarly, the second-order moment can be bounded by

$$\mathbb{E}[\|e_2\|^2] \leq \mathbb{E}\left[\frac{1}{\lambda^2}\|\nabla f_i(w) - \nabla\tilde{f}_i(w,\mathcal{D}_1)\|^2\right] \leq \frac{1}{\lambda^2|\mathcal{D}_1|}\sigma_G. \tag{16}$$

Combining (14), (15) and (16), we have

$$\|\mathbb{E}[g_i(w) - \nabla F_i(w)]\| \leq \lambda\|(I - \frac{1}{\lambda}\nabla^2 f_i(w))\mathbb{E}[e_2]\| \leq \frac{2}{\sqrt{|\mathcal{D}_1|}}\sqrt{1 - \frac{|\mathcal{D}_1|-1}{|\mathcal{D}|-1}}\sigma_G.$$

Furthermore, it holds

$$\mathbb{E}\left[\|g_i(w) - \nabla F_i(w)\|^2\right]$$

$$\leq 3\lambda^2\left(\mathbb{E}\left[\|e_1\|^2\|(w - \text{prox}_{\frac{1}{\lambda}h}(w - \frac{1}{\lambda}\nabla f_i(w)))\|^2 + \|I - \frac{1}{\lambda}\nabla^2 f_i(w)\|^2\|e_2\|^2 + \|e_1\|^2\|e_2^2\|\right]\right)$$

$$\leq 3\lambda^2\left(\frac{1}{|\mathcal{D}_2|\lambda^2}\sigma_H^2 \cdot \frac{4B^2}{\lambda^2} + \left(1 + \frac{L}{\lambda}\right)^2 \cdot \frac{1}{|\mathcal{D}_1|\lambda^2}\sigma_G^2 + \frac{1}{|\mathcal{D}_2|\lambda^2}\sigma_H^2 \cdot \frac{1}{|\mathcal{D}_1|\lambda^2}\sigma_G^2\right)$$

$$= \frac{12}{|\mathcal{D}_1|}\sigma_G^2 + \frac{12B^2}{|\mathcal{D}_2|\lambda^2}\sigma_H^2 + \frac{3}{|\mathcal{D}_1||\mathcal{D}_2|\lambda^2}\sigma_G^2\sigma_H^2,$$

where the first equality is from the Cauchy-Schwarz inequality, the second inequality is from $\|w - \text{prox}_{\frac{1}{\lambda}h}(w - \frac{1}{\lambda}\nabla f(w))\| \leq \frac{2B}{\lambda}$, (14) and (16). We complete the proof.

A.4 PROOF OF LEMMA 4

Note that the local update of Algorithm 1 is

$$w_{k,t+1}^i = w_{k,t} - \eta g_i(w_{k,t}^i), \tag{17}$$

where $g_i(w_{k,t}^i)$ is the estimated gradient of $F_i$ at $w_{k,t}^i$. Define $C_t := \frac{1}{N} \sum_{i=1}^N \mathbb{E}\left[\left\|w_{k,t}^i - w_{k,t}\right\|\right]$.
We have $S_0 = 0$ since $w_{k,0}^i = w_k$ for any $i$. It follows from the local update scheme (17) that

$$
\begin{aligned}
C_{t+1} &= \frac{1}{N} \sum_{i=1}^N \mathbb{E}\left[\left\|w_{k,t+1}^i - w_{k,t+1}\right\|\right] \\
&= \frac{1}{N} \sum_{i=1}^n \mathbb{E}\left[\left\|w_{k,t}^i - \eta g_i\left(w_{k,t}^i\right) - \frac{1}{N} \sum_{j=1}^N \left(w_{k,t}^j - \eta g_j\left(w_{k,t}^j\right)\right)\right\|\right] \\
&\le C_t + \eta \frac{1}{N} \sum_{i=1}^N \underbrace{\mathbb{E}\left[\left\|g_i\left(w_{k,t}^i\right) - \frac{1}{N} \sum_{j=1}^N g_j\left(w_{k,t}^j\right)\right\|\right]}_{=:b_1}.
\end{aligned}
\tag{18}
$$

For $b_1$, it holds

$$
\begin{aligned}
b_1 &\le \frac{\eta}{N} \sum_{i=1}^N \mathbb{E}\left[\left\|\nabla F_i\left(w_{k,t}^i\right) - \frac{1}{N} \sum_{j=1}^N \nabla F_j\left(w_{k,t}^j\right)\right\|\right] + \frac{\eta}{N} \sum_{i=1}^N \mathbb{E}\left[\left\|\nabla F_i\left(w_{k,t}^i\right) - g_i\left(w_{k,t}^i\right)\right\|\right] \\
&\quad + \frac{\eta}{N} \sum_{i=1}^N \mathbb{E}\left[\frac{1}{N} \sum_{j=1}^N \left\|\nabla F_j\left(w_{k,t}^j\right) - g_j\left(w_{k,t}^j\right)\right\|\right] \\
&\le \frac{\eta}{N} \sum_{i=1}^N \mathbb{E}\left[\left\|\nabla F_i\left(w_{k,t}^i\right) - \frac{1}{N} \sum_{j=1}^N \nabla F_j\left(w_{k,t}^j\right)\right\|\right] + 2\eta\sigma_F
\end{aligned}
\tag{19}
$$

where the first inequality is due to the triangle inequality and the last inequality is from Lemma 3.
Combining (18) and (19), and defining $\alpha_i := \nabla F_i\left(w_{k,t}^i\right) - \nabla F_i(w_{k,t})$, we have

$$
\begin{aligned}
C_{t+1} &\le C_t + 2\eta\sigma_F + \frac{\eta}{N} \sum_{i=1}^N \mathbb{E}\left[\left\|\nabla F_i\left(w_{k,t}^i\right) - \frac{1}{N} \sum_{j=1}^N \nabla F_j\left(w_{k,t}^j\right)\right\|\right] \\
&= C_t + 2\eta\sigma_F + \frac{\eta}{N} \sum_{i=1}^N \mathbb{E}\left[\left\|\nabla F_i(w_{k,t}) - \frac{1}{N} \sum_{j=1}^N \nabla F_j(w_{k,t})\right\|\right] \\
&\quad + \frac{\eta}{N} \sum_{i=1}^N \mathbb{E}\left[\left\|\alpha_i - \frac{1}{N} \sum_{j=1}^N \alpha_j\right\|\right].
\end{aligned}
\tag{20}
$$

It follows from Lemma 1 that $\|\alpha_i\| \le L_F \left\|w_{k,t}^i - w_{k,t}\right\|$. Consequently,

$$\frac{1}{N} \sum_{i=1}^N \mathbb{E}\left[\|\alpha_i\|\right] \le L_F C_t.$$

Plugging the above inequality into (20) leads to

$$
\begin{aligned}
C_{t+1} &\le (1 + 2\eta L_F) C_t + 2\eta\sigma_F + \frac{\eta}{N} \sum_{i=1}^N \mathbb{E}\left[\left\|\nabla F_i(w_{k,t}) - \frac{1}{N} \sum_{j=1}^N \nabla F_j(w_{k,t})\right\|\right] \\
&\le (1 + 2\eta L_F) C_t + 2\eta\left(\sigma_F + \gamma_F\right)
\end{aligned}
\tag{21}
$$

where the last inequality is due to Lemma 2. From the recursion (21), we have

$$
\begin{aligned}
C_{t+1} &\le \left( \sum_{j=0}^{t} (1 + 2\eta L_F)^j \right) 2\eta \left( \sigma_F + \gamma_F \right) \\
&\le 2\eta(t+1) \left( 1 + 2\eta L_F \right)^t \left( \sigma_F + \gamma_F \right) \\
&\le 2\eta(t+1)(1 + \frac{1}{5R})^t (\sigma_F + \gamma_F) \le 4\eta(t+1)(\sigma_F + \gamma_F),
\end{aligned}
\tag{22}
$$

which the third inequality is from $\eta \le \frac{1}{10 L_F R}$ and the last inequality is due to $(1 + \frac{1}{5R})^t \le e^{\frac{1}{5}} < 2$. We finish the proof of (10). For the proof of (11), let us define $D_t := \frac{1}{N} \sum_{i=1}^{N} \mathbb{E} \left[ \left\| w_{k,t}^i - w_{k,t} \right\|^2 \right]$. Since $w_{k,t}^0 = w_k$, $\forall i = 1, \ldots, N$, it holds $D_0 = 0$. Following (17), we have

$$
\begin{aligned}
D_{t+1} &= \frac{1}{N} \sum_{i=1}^{N} \mathbb{E} \left[ \left\| w_{k,t+1}^i - w_{k,t+1} \right\|^2 \right] \\
&= \frac{1}{N} \sum_{i=1}^{N} \mathbb{E} \left[ \left\| w_{k,t}^i - \eta g_i \left( w_{k,t}^i \right) - \frac{1}{N} \sum_{j=1}^{N} \left( w_{k,t}^j - \eta g_j \left( w_{k,t}^j \right) \right) \right\|^2 \right] \\
&\le \frac{1+\nu}{N} \sum_{i=1}^{N} \mathbb{E} \left[ \left\| w_{k,t}^i - \frac{1}{N} \sum_{j=1}^{N} w_{k,t}^j \right\|^2 \right] \\
&\quad + \eta^2 \frac{1 + 1/\nu}{N} \sum_{i=1}^{N} \mathbb{E} \left[ \left\| g_i \left( w_{k,t}^i \right) - \frac{1}{N} \sum_{j=1}^{N} g_j \left( w_{k,t}^j \right) \right\|^2 \right] \\
&\le (1+\nu) D_t + \underbrace{\eta^2 \frac{1 + 1/\nu}{N} \sum_{i=1}^{N} \mathbb{E} \left[ \left\| g_i \left( w_{k,t}^i \right) - \frac{1}{N} \sum_{j=1}^{N} g_j \left( w_{k,t}^j \right) \right\|^2 \right]}_{=:b_2},
\end{aligned}
\tag{23}
$$

where the first equality is from $\|a+b\|^2 \le (1+\phi)\|a\|^2 + (1+1/\phi)\|b\|^2$ for any $\nu > 0$. For $b_2$, it holds that

$$
\begin{aligned}
b_2 &\le 2\eta^2 \frac{1 + 1/\nu}{N} \sum_{i=1}^{N} \left[ \mathbb{E} \left[ \left\| \nabla F_i \left( w_{k,t}^i \right) - \frac{1}{n} \sum_{j=1}^{n} \nabla F_j \left( w_{k,t}^j \right) \right\|^2 \right] \right. \\
&\quad \left. + 2\mathbb{E} \left[ \left\| \left( g_i \left( w_{k,t}^i \right) - \nabla F_i \left( w_{k,t}^i \right) \right) + \frac{1}{N} \sum_{j=1}^{N} \left( \nabla F_j \left( w_{k,t}^j \right) - g_j \left( w_{k,t}^j \right) \right) \right\|^2 \right] \right] \\
&\le 2\eta^2 \frac{1 + 1/\nu}{N} \sum_{i=1}^{N} \left[ \mathbb{E} \left[ \left\| \nabla F_i \left( w_{k,t}^i \right) - \frac{1}{n} \sum_{j=1}^{n} \nabla F_j \left( w_{k,t}^j \right) \right\|^2 \right] \right. \\
&\quad \left. + 4\mathbb{E} \left[ \left\| g_i \left( w_{k,t}^i \right) - \nabla F_i \left( w_{k,t}^i \right) \right\|^2 + \frac{1}{N} \sum_{j=1}^{N} \left\| \nabla F_j \left( w_{k,t}^j \right) - g_j \left( w_{k,t}^j \right) \right\|^2 \right] \right] \\
&\le 4\eta^2 \frac{1 + 1/\nu}{N} \sum_{i=1}^{N} \left[ \mathbb{E} \left[ \left\| \nabla F_i \left( w_{k,t} \right) - \frac{1}{N} \sum_{j=1}^{N} \nabla F_j \left( w_{k,t} \right) \right\|^2 + \mathbb{E} \left[ \left\| \alpha_i - \frac{1}{N} \sum_{j=1}^{N} \alpha_j \right\|^2 \right] \right] \right] + 4\sigma_F^2
\end{aligned}
$$

$$\leq 4\eta^2 \frac{1+1/\nu}{N} \sum_{i=1}^{N} \left[ \gamma_F^2 + 2L_F^2 \left( \mathbb{E}\left[ \|w_{k,t}^i - w_{k,t}\|^2 \right] + \mathbb{E}\left[ \frac{1}{N} \sum_{j=1}^{N} \|w_{k,t}^j - w_{k,t}\|^2 \right] \right) + 4\sigma_F^2 \right]$$

$$\leq 4\eta^2 (1+1/\nu) \left[ \gamma_F^2 + 4L_F^2 D_t + 4\sigma_F^2 \right], \tag{24}$$

where the first inequality the second inequality are due to the Cauchy–Schwarz inequality, the third inequality is from Lemma 3, the Cauchy–Schwarz inequality and $\alpha_i = \nabla F_i(w_{k,t}^i) - \nabla F_i(w_{k,t})$, the fourth inequality is from Lemma 1, and the last inequality is obtained using the definition of $D_t$. Plugging (24) into (23) yields

$$D_{t+1} \leq (1+\nu)D_t + 4\eta^2(1+1/\nu)\left[ \gamma_F^2 + 4L_F^2 D_t + 4\sigma_F^2 \right]$$

$$\leq \left( \sum_{j=0}^{t} (1+\nu + 16(1+1/\nu)\eta^2 L_F^2)^j \right) \cdot 4\eta^2(1+1/\nu)(\gamma_F^2 + 4\sigma_F^2) \tag{25}$$

$$\leq 4\eta^2(t+1)(1+\nu + 16(1+1/\nu)\eta^2 L_F^2)^t(1+1/\nu)(\gamma_F^2 + 4\sigma_F^2).$$

Taking $\nu = \frac{1}{2R}$ and $\eta \leq \frac{1}{10L_F R}$ give

$$(1+\nu + 16(1+1/\nu)\eta^2 L_F^2)^t = \left( 1 + \frac{1}{2R} + 16(1+2R)\eta^2 L_F^2 \right)^t$$

$$\leq \left( 1 + \frac{1}{2R} + 16(1+2R)\frac{1}{100R^2} \right)^t \leq \left( 1 + \frac{1}{R} \right)^t \leq 4,$$

where the first inequality is due to $\eta <\leq \frac{1}{10L_F R}$, the second inequality is from $1 + 2R \leq 3R$, and the last inequality is based on the fact $(1 + \frac{1}{R})^R \leq e$. Plugging the above inequality into (25), we have

$$D_{t+1} \leq 4\eta^2 \cdot (t+1) \cdot 4 \cdot 3R(\gamma_F^2 + 4\sigma_F^2) \leq 48(t+1)R\eta^2(\gamma_F^2 + 4\sigma_F^2).$$

We complete the proof.

## A.5 PROOF OF THEOREM 1

Denote by $\mathcal{F}_k^t$ the $\sigma$-field generated by $\left\{ w_{k,t}^i \right\}_{i=1}^{N}$. Define the averaged iterate $\bar{w}_{k,t} := \frac{1}{S} \sum_{i \in \mathcal{S}_k} w_{k,t}^i$. Then,

$$\bar{w}_{k+1,t+1} = \frac{1}{S} \sum_{i \in \mathcal{S}_k} \left( w_{k+1,t}^i - \eta g_i(w_{k+1,t}^i) \right) = \bar{w}_{k+1,t} - \frac{\eta}{S} \sum_{i \in \mathcal{S}_k} g_i(w_{k+1,t}^i). \tag{26}$$

From the Lipschitz continuous property of the gradient of $F_i$ 1, we have

$$F\left( \bar{w}_{k+1,t+1} \right)$$

$$\leq F\left( \bar{w}_{k+1,t} \right) + \nabla F\left( \bar{w}_{k+1,t} \right)^\top \left( \bar{w}_{k+1,t+1} - \bar{w}_{k+1,t} \right) + \frac{L_F}{2} \left\| \bar{w}_{k+1,t+1} - \bar{w}_{k+1,t} \right\|^2$$

$$= F\left( \bar{w}_{k+1,t} \right) - \eta \nabla F\left( \bar{w}_{k+1,t} \right)^\top \left( \frac{1}{S} \sum_{i \in \mathcal{S}_k} g_i\left( w_{k+1,t}^i \right) \right) + \frac{L_F}{2}\eta^2 \left\| \frac{1}{S} \sum_{i \in \mathcal{S}_k} g_i\left( w_{k+1,t}^i \right) \right\|^2 \tag{27}$$

where the inequality is from the Lipschitz gradient property of $F$ and the equality is due to (26). By taking expectation on (27), we obtain

$$\mathbb{E}\left[ F\left( \bar{w}_{k+1,t+1} \right) \right] \leq \mathbb{E}\left[ F\left( \bar{w}_{k+1,t} \right) \right] \underbrace{- \eta \mathbb{E}\left[ \nabla F\left( \bar{w}_{k+1,t} \right)^\top \left( \frac{1}{S} \sum_{i \in \mathcal{S}_k} g_i\left( w_{k+1,t}^i \right) \right) \right]}_{=:q_1}$$

$$\underbrace{+ \frac{L_F}{2}\eta^2 \mathbb{E}\left[ \left\| \frac{1}{S} \sum_{i \in \mathcal{S}_k} g_i(w_{k+1,t}^i) \right\|^2 \right]}_{=:q_2}. \tag{28}$$

The sketch of the proof is to estimate the difference between the stochastic gradient $\frac{1}{S} \sum_{i \in \mathcal{S}_k} g_i \left( w_{k+1,t}^i \right)$ and $\nabla F(\bar{w}_{k+1,t})$ and derive a decrease of $F$ with respect to $\|\nabla F(\bar{w}_{k+1,t})\|^2$. Once one-step decrease is obtained, the complexity result is obtained by summing over all iterates. Firstly, we use the following split on $g_i(w_{k+1,t}^i)$, namely,

$$\frac{1}{S} \sum_{i \in \mathcal{S}_k} g_i \left( w_{k+1,t}^i \right) = X + Y + Z + Q + \nabla F(\bar{w}_{k+1,t}), \tag{29}$$

where

$$X = \frac{1}{S} \sum_{i \in \mathcal{S}_k} \left( g_i \left( w_{k+1,t}^i \right) - \nabla F_i \left( w_{k+1,t}^i \right) \right),$$

$$Y = \frac{1}{S} \sum_{i \in \mathcal{S}_k} \left( \nabla F_i \left( w_{k+1,t}^i \right) - \nabla F_i \left( w_{k+1,t} \right) \right),$$

$$Z = \frac{1}{S} \sum_{i \in \mathcal{S}_k} \left( \nabla F_i \left( w_{k+1,t} \right) - \nabla F_i \left( \bar{w}_{k+1,t} \right) \right),$$

$$Q = \frac{1}{S} \sum_{i \in \mathcal{S}_k} \nabla F_i \left( \bar{w}_{k+1,t} \right) - \nabla F(\bar{w}_{k+1,t}).$$

Next, we bound the moments of $X, Y, Z$ and $Q$.

- It follows from the Cauchy-Schwarz inequality that

$$\|X\|^2 \leq \frac{1}{S} \sum_{i \in \mathcal{S}_k} \left\| g_i \left( w_{k+1,t}^i \right) - \nabla F_i \left( w_{k+1,t}^i \right) \right\|^2.$$

By the tower rule, we have

$$\mathbb{E}\left[ \|X\|^2 \right] = \mathbb{E}\left[ \mathbb{E}\left[ \|X\|^2 | \mathcal{F}_{k+1}^t \right] \right] \leq \sigma_F^2. \tag{30}$$

- Note that

$$\begin{aligned} \|Y\|^2 &\leq \frac{1}{S} \sum_{i \in \mathcal{S}_k} \left\| \nabla F_i \left( w_{k+1,t}^i \right) - \nabla F_i \left( w_{k+1,t} \right) \right\|^2 \\ &\leq \frac{L_F^2}{S} \sum_{i \in \mathcal{S}_k} \left\| w_{k+1,t}^i - w_{k+1,t} \right\|^2 \\ &\leq 48 R(R-1) \eta^2 L_F^2 (\gamma_F^2 + 4\sigma_F^2), \end{aligned} \tag{31}$$

where the first inequality is from the Cauchy-Schwarz inequality, the second inequality is due to the Lipschitz gradient of $F_i$ given in Lemma 1, and the last inequality is based on (11) and $t \leq R - 1$.

- Using the variance of sampling without replacement, we have

$$\begin{aligned} \mathbb{E}\left[ \|\bar{w}_{k+1,t} - w_{k+1,t}\|^2 | \mathcal{F}_{k+1}^t \right] &\leq \mathbb{E}\left[ \frac{1}{S} \sum_{i \in \mathcal{S}_k} \|w_{k+1,t}^i - w_{k+1,t}\|^2 \right] \\ &= \frac{1}{SN} \sum_{i=1}^{N} \|w_{k+1,t}^i - w_{k+1,t}\|^2 \cdot \left( 1 - \frac{S-1}{N-1} \right). \end{aligned} \tag{32}$$

By the gradient Lipschitz property of $F_i$ given in Lemma 1, we obtain

$$\begin{aligned} \mathbb{E}\left[ \|Z\|^2 \right] &\leq \mathbb{E}\left[ \mathbb{E}\left[ \frac{L_F^2}{S} \sum_{i \in \mathcal{S}_k} \|\bar{w}_{k+1,t} - w_{k+1,t}\|^2 | \mathcal{F}_{k+1}^t \right] \right] \\ &\leq \frac{L_F^2}{SN} \sum_{i=1}^{N} \|w_{k+1,t}^i - w_{k+1,t}\|^2 \cdot \left( 1 - \frac{S-1}{N-1} \right) \\ &\leq \frac{48(N-S)R(R-1)\eta^2 L_F^2 (\gamma_F^2 + 4\sigma_F^2)}{S(N-1)}, \end{aligned} \tag{33}$$

where the second inequality is due to (32) and the last inequality is from (11).

- From Lemma 2 and similar derivations as (32), we have

$$\mathbb{E}\left[\|Q\|^2\right] \le \mathbb{E}\left[\mathbb{E}\left[\frac{1}{S}\sum_{i\in\mathcal{S}_k}\|\nabla F_i(\bar{w}_{k+1,t}) - \nabla F(\bar{w}_{k+1,t})\|^2|\mathcal{F}_{k+1}^t\right]\right] \le \frac{N-S}{S(N-1)}\gamma_F^2. \quad (34)$$

In addition, using the tower rule, it holds that

$$\mathbb{E}[Q] = \mathbb{E}\left[\mathbb{E}\left[\nabla F_i(\bar{w}_{k+1,t}) - \nabla F(\bar{w}_{k+1,t})|\mathcal{F}_{k+1}^t\right]\right] = 0. \quad (35)$$

With the above estimates, we have

$$
\begin{aligned}
q_1 \le &\eta\mathbb{E}\left[\nabla F(\bar{w}_{k+1,t})^\top (X+Y+Z+Q+\nabla F(\bar{w}_{k+1,t}))\right] \\
\ge &\eta\mathbb{E}\left[\nabla F(\bar{w}_{k+1,t})^\top (Q+\nabla F(\bar{w}_{k+1,t}))\right] - \eta\left|\mathbb{E}\left[\nabla F(\bar{w}_{k+1,t})^\top X\right]\right| \\
&- \frac{\eta}{4}\mathbb{E}\left[\|\nabla F(\bar{w}_{k+1,t})\|^2\right] - \eta\mathbb{E}[\|Y+Z\|^2], \\
= &\eta\mathbb{E}\left[\|\nabla F(\bar{w}_{k+1,t})\|^2\right] - \eta\left|\mathbb{E}\left[\nabla F(\bar{w}_{k+1,t})^\top X\right]\right| - \frac{\eta}{4}\mathbb{E}\left[\|\nabla F(\bar{w}_{k+1,t})\|^2\right] - \eta\mathbb{E}[\|Y+Z\|^2],
\end{aligned}
$$
$$(36)$$

where the second inequality is due to the Cauchy-Schwarz inequality and the last equality is from (35). For the second term in (36), we have

$$
\begin{aligned}
\left\|\mathbb{E}\left[\nabla F(\bar{w}_{k+1,t})^\top X\right]\right\| &= \left\|\mathbb{E}\left[\mathbb{E}\left[\nabla F(\bar{w}_{k+1,t})^\top X \mid \mathcal{F}_{k+1}^t\right]\right]\right\| \\
&= \left\|\mathbb{E}\left[\nabla F(\bar{w}_{k+1,t})^\top \mathbb{E}\left[X \mid \mathcal{F}_{k+1}^t\right]\right]\right\| \\
&\le \frac{1}{4}\mathbb{E}\left[\|\nabla F(\bar{w}_{k+1,t})\|^2\right] + \mathbb{E}\left[\|\mathbb{E}\left[X \mid \mathcal{F}_{k+1}^t\right]\|^2\right] \\
&\le \frac{1}{4}\mathbb{E}\left[\|\nabla F(\bar{w}_{k+1,t})\|^2\right] + 4r\sigma_G^2,
\end{aligned}
$$
$$(37)$$

where $r := \max_{k,t,i}\left(1 - \frac{|\mathcal{D}_{k,t}^i|-1}{|\mathcal{D}^i|-1}\right)/|\mathcal{D}_{k,t}^i|$ in the last inequality follows from Lemma 3. For the last term of (36), it holds that

$$
\begin{aligned}
\mathbb{E}\left[\|Y+Z\|^2\right] &\le 2\left(\mathbb{E}\left[\|Y\|^2\right] + \mathbb{E}\left[\|Z\|^2\right]\right) \\
&\le 96R(R-1)\eta^2 L_F^2(\gamma_F^2 + 4\sigma_F^2)\left(1 + \frac{N-S}{S(N-1)}\right) \\
&\le 192R(R-1)\eta^2 L_F^2(\gamma_F^2 + 4\sigma_F^2),
\end{aligned}
$$
$$(38)$$

where the first inequality is from the Cauchy-Schwarz inequality, the second inequality due to (31) and (33). Plugging (37) and (38) in (36) yields

$$q_1 \ge \frac{\eta}{2}\mathbb{E}\left[\|\nabla F(\bar{w}_{k+1,t})\|^2\right] - 192R(R-1)\eta^3 L_F^2(\gamma_F^2 + 4\sigma_F^2) - 4r\eta\sigma_G^2. \quad (39)$$

The remaining term in (28) needs to be bounded is $q_2$. It follows from the Cauchy-Schwarz inequality that

$$
\begin{aligned}
\left\|\frac{1}{S}\sum_{i\in\mathcal{S}_k}g_i\left(w_{k+1,t}^i\right)\right\|^2 &\le 2\|X+Y+Z\|^2 + 2\|Q+\nabla F(\bar{w}_{k+1,t})\|^2 \\
&\le 4(\|X\|^2 + \|Y+Z\|^2) + 4(\|Q\|^2 + \|\nabla F(\bar{w}_{k+1,t})\|^2).
\end{aligned}
$$
$$(40)$$

By (30), (38), (34) and (40), we have

$$
\begin{aligned}
q_2 \le &2L_F\eta^2\left(\sigma_F^2 + 192R(R-1)\eta^2 L_F^2(\gamma_F^2 + 4\sigma_F^2) + \frac{N-S}{S(N-1)}\gamma_F^2 + \|\nabla F(\bar{w}_{k+1,t})\|^2\right) \\
\le &2L_F\eta^2\|\nabla F(\bar{w}_{k+1,t})\|^2 + 400L_F^3\eta^4 R(R-1)(\gamma_F^2 + 4\sigma_F^2) + 2L_F\eta^2\left(\frac{N-S}{S(N-1)}\gamma_F^2 + \sigma_F^2\right).
\end{aligned}
$$
$$(41)$$

Plugging (36) and (41) into (28) gives

$$
\begin{aligned}
&\mathbb{E}\left[F\left(\bar{w}_{k+1,t+1}\right)\right] \\
&\leq \mathbb{E}\left[F\left(\bar{w}_{k+1,t}\right)\right] - \eta\left(1/2 - 2\eta L_F\right)\mathbb{E}\left[\left\|\nabla F\left(\bar{w}_{k+1,t}\right)\right\|^2\right] \\
&\quad + 200\left(1 + 2\eta L_F\right)\eta^3 L_F^2 R(R-1)\left(\gamma_F^2 + 4\sigma_F^2\right) + 2L_F\eta^2\left(\frac{N-S}{S(N-1)}\gamma_F^2 + \sigma_F^2\right) + 4r\eta\sigma_G^2 \\
&\leq \mathbb{E}\left[F\left(\bar{w}_{k+1,t}\right)\right] - \frac{\eta}{4}\mathbb{E}\left[\left\|\nabla F\left(\bar{w}_{k+1,t}\right)\right\|^2\right] + 400\eta^3 R^2(\gamma_F^2 + 4\sigma_F^2) \\
&\quad + 2L_F\eta^2\left(\frac{N-S}{S(N-1)}\gamma_F^2 + \sigma_F^2\right) + 4r\eta\sigma_G^2,
\end{aligned}
$$

(42)

where the last inequality is due to $\eta \leq \frac{1}{10RL_F}$. Note that $\bar{w}_{k,R} = w_k$. Summing over (42) for all $t = 0, \ldots, R-1$ and $k = 0, \ldots, K-1$ yields

$$
\begin{aligned}
\mathbb{E}\left[F\left(w_K\right)\right] \leq &F\left(w_0\right) - \frac{\eta RK}{4}\left(\frac{1}{RK}\sum_{k=0}^{K-1}\sum_{t=0}^{R-1}E\left[\left\|\nabla F\left(\bar{w}_{k+1,t}\right)\right\|^2\right]\right) + 400\eta^3 R^3 K(\gamma_F^2 + 4\sigma_F^2) \\
&+ 2RKL_F\eta^2\left(\frac{N-S}{S(N-1)}\gamma_F^2 + \sigma_F^2\right) + 4RKr\eta\sigma_G^2.
\end{aligned}
$$

Therefore,

$$
\begin{aligned}
\frac{1}{RK}\sum_{k=0}^{K-1}\sum_{t=0}^{R-1}E\left[\left\|\nabla F\left(\bar{w}_{k+1,t}\right)\right\|^2\right] \leq &\frac{4}{\eta RK}\left(F\left(w_0\right) - \mathbb{E}\left[F\left(w_K\right)\right] + \eta RK\sigma_T^2\right) \\
\leq &\frac{4\left(F\left(w_0\right) - F^*\right)}{\eta RK} + 1600\eta^2 R^2(\gamma_F^2 + 4\sigma_F^2) \\
&+ 8L_F\eta\left(\frac{N-S}{S(N-1)}\gamma_F^2 + \sigma_F^2\right) + 16r\sigma_G^2,
\end{aligned}
$$

which $F^*$ the minimal value of $F$. This completes the proof.

