# OpenReview forum: "Personalized federated composite learning with forward-backward envelopes"
_ICLR.cc/2023/Conference — Submitted to ICLR 2023_

### Official Review · Reviewer_WBRe · 2022-10-24

**Confidence:** 4
**Correctness:** 3
**Technical Novelty And Significance:** 2
**Empirical Novelty And Significance:** 2
**Recommendation:** 5

**Clarity, Quality, Novelty And Reproducibility:**

The writting of the paper should be improved; novelty is maringal; reproducibility is unclear;

**Strength And Weaknesses:**

Pros:
- The story line of the paper is clear and related works are well-addressed to my knowledge.

Cons:
- The presentation of this paper should be improved: (1) the author(s) should use \citep in many places, for example in the first paragraph. (ii) Figure 1,2,3,4,5,6 are not properly cropped. (3) Notations are kind of heavy, it is better to include a subsection for notations.
- From the perspective of personalized FL, the paper seems to be a direct extension of [1] to composite problems. However, generalizing [1] to the composite problem does not seem to be a challenging task.
- The author(s) spent much space for the theoretical analysis (page 5 and page 6). However, Theorem 1 does not seem to be a particularly strong results, the algorithm uses second order information and does not coverge faster.
- Using second order information (the Hessian matrix) is expensive in practice, this issue is only briefly mentioned above Section 3. It is better to discuss more on how to efficiently leverage the second order information, i.e., how to do the Hessian-vector product efficiently and what is the time complexity of the proposed algorithm.
- Experiments are also not very convincing, the datasets are mostly aftifically generated, the only real dataset is the MNIST dataset. It is more convincing the evaluate the proposed algorithm on more real-world datasets.


[1] Canh T Dinh, Nguyen Tran, and Josh Nguyen. Personalized federated learning with moreau envelopes. Advances in Neural Information Processing Systems, 33:21394–21405, 2020.

**Summary Of The Paper:**

The author(s) extends the Moreu-envelop based personalized FL to the composite problems. Convergence analysis under certain assumptions are developed. Experiments on synethtic data and MNIST are conducted to evaluate the proposed algorithm.

**Summary Of The Review:**

The presentation of the paper should be improved. On the theory side, the contribution of the paper is kind of marginal compared with the exiting Moreu-envelop based personalized FL; on the empirical side, the experiments are mostly on synthetic and tiny datasets. The author(s) should consider improving either from the theory or the empirical aspect.

---

> ### Author Response · Authors · 2022-11-19
> **Response to Reviewer WBRe**
>
> Thank you for the valuable feedbacks. We have revised our paper accordingly. All the updates are in blue color.  We address the questions as follows.
>
> 1. Q: "The presentation of this paper should be improved: (1) the author(s) should use ``citep'' in many places, for example in the first paragraph. (2) Figure 1,2,3,4,5,6 are not properly cropped. (3) Notations are kind of heavy, it is better to include a subsection for notations."
>
>    A: (1) All ``cite'' has been changed to ``citep''. (2) The sizes of the figures are due to the tight mode of the plt.savefig. We keep them due to their economic sizes.  (3) We have added the notation part.
>
> 2. Q: "From the perspective of personalized FL, the paper seems to be a direct extension of [1] to composite problems. However, generalizing [1] to the composite problem does not seem to be a challenging task. The author(s) spent much space for the theoretical analysis (page 5 and page 6)."
>
>     A: The main goal of this paper is to derive a personalized model for FCO, which has not been considered yet. By taking advantage of FBE, the federated smooth optimization with explicit form of  gradients is constructed, where both federated parameters and personalized parameters can be obtained. Our main contribution in the theoretic convergence is to translate the assumptions on FBE required by FedAvg to the original functions $f_i$ and $h$.
>
> 3. Q: "Using second order information (the Hessian matrix) is expensive in practice, this issue is only briefly mentioned above Section 3. It is better to discuss more on how to efficiently leverage the second order information, i.e., how to do the Hessian-vector product efficiently and what is the time complexity of the proposed algorithm."
>
>     A: The gradient of FBE involves an additional Hessian-vector product than the proximal gradient. But the additional computation is not much because it could be calculated through numerical differences between two gradients, see [2]. Specifically, the numerical approximation is based on $\nabla^2 f(x) u \approx (\nabla f(x+tu) - \nabla f(x)) /t$ with small $t > 0$. We note that the proximal gradient step (i.e., ignore the Hessian part) can also serve as an inexact gradient to reduce the computations. Similar ideas are conducted in the personalized FedAvg paper [2].
>
>
> 4. Q: "Experiments are also not very convincing, the datasets are mostly aftifically generated, the only real dataset is the MNIST dataset. It is more convincing the evaluate the proposed algorithm on more real-world datasets."
>
>     A: We do not put the results due to the page limit.
>
> [1] Canh T Dinh, Nguyen Tran, and Josh Nguyen. Personalized federated learning with moreau envelopes. Advances in Neural Information Processing Systems, 33:21394–21405, 2020.
>
> [2] Fallah, Alireza, Aryan Mokhtari, and Asuman Ozdaglar. "Personalized federated learning: A meta-learning approach." arXiv preprint arXiv:2002.07948 (2020).

---

### Official Review · Reviewer_4Vq3 · 2022-10-25

**Confidence:** 4
**Correctness:** 3
**Technical Novelty And Significance:** 2
**Empirical Novelty And Significance:** 2
**Recommendation:** 3

**Clarity, Quality, Novelty And Reproducibility:**

The writing can significantly be improved.
The novelty of this paper is limited.  This paper basically follows the existing methods and
theoretical analysis such as in (https://arxiv.org/pdf/2011.08474.pdf).

**Strength And Weaknesses:**

Strength:

This paper proposed a personalization federated learning method (i.e., pFedFBE) by using forward-backward envelope as clients’ loss functions. It provided the convergence analysis of the proposed method, which shows the same convergence complexity results as FedAvg for FL with unconstrained smooth objectives.

Weakness:

The novelty of this paper is limited. This paper basically follows the existing methods and  theoretical analysis such as in (https://arxiv.org/pdf/2011.08474.pdf).

**Summary Of The Paper:**

This paper proposed a personalization federated learning method (i.e., pFedFBE) by using forward-backward envelope as clients’ loss functions. It provided the convergence analysis  of the proposed method, which shows the same convergence complexity results as FedAvg for FL with unconstrained smooth objectives. Numerical experimental results demonstrate the effectiveness of the proposed pFedFBE method.

**Summary Of The Review:**

This paper proposed a personalization federated learning method (i.e., pFedFBE)  by using forward-backward envelope as clients’ loss functions. It provided the convergence analysis  of the proposed method, which shows the same convergence complexity results as FedAvg for FL with unconstrained smooth objectives. Numerical experimental results demonstrate the effectiveness of the proposed pFedFBE method.

Some Comments:

1. Assumption 1-(A3) is stronger than the basic assumptions in the existing FL methods such as FedAvg.

2. It would be great if the authors would detail the tuning parameters in the experiments.


------------------------------------------------------------------------------------------------------------------------------------------
--------------------------------------------------------------------------------------------------------------------------------------------

The authors still did not solve my main concern: the limited novelty of this paper. So I support to reject this paper.

---

> ### Author Response · Authors · 2022-11-19
> **Response to Reviewer 4Vq3**
>
> Thank you for the valuable feedbacks. We have revised our paper accordingly. All the updates are in blue color.  We address the questions as follows.
>
> 1. Q: "The novelty of this paper is limited. This paper basically follows the existing methods and theoretical analysis such as in (https://arxiv.org/pdf/2011.08474.pdf)."
>
>     A: The listed arxiv paper directly solves the federated nonsmooth optimization problem and only can obtain the federations, while we are foucsing on a Federated smooth optimization problem induced by FBE and allows both federations and personalizations under certain conditions. In the theoretical analysis, our sketch is to establish the lower boundedness, the gradient Lipschitz continuity, the bounded variance, and the bounded divergence of FBE based on proper assumptions on $f_i$ and $h$. Then, the convergence of the federate iterate  can be proved as the classic FedAvg method.
>
> 2. Q: "Assumption (A1)-(A3) is stronger than the basic assumptions in the existing FL methods such as FedAvg."
>
>     A: When comes to the nonsmooth setting, FedAvg (the gradient should be replaced by the subgradient) will also need the Lipschitz continuity of the nonsmooth term $h$. The boundedness of the second-order derivatives can be guaranteed if the function itself is of gradient Lipschitz. The Lipschitz continuity of the Hessian does not mean the function itself is third-order differentiable. It is not a strong condition and covers many applications, e.g., Lasso, matrix completions, and logistic regressions. The Lispchitz continuity assumption of the nonsmooth term, i.e., the boundedness of subgradients, is also standard in the field of decentralized optimization [1], which is used to bound the consensus error. The $\ell_1$ norm we use in the experiments is with bounded subgradients.
>
>
> 3. Q: "It would be great if the authors would detail the tuning parameters in the experiments."
>
>     A: We have added the specific values of tuning parameters accordingly.
>
> [1] Zeng, Jinshan, and Wotao Yin. "On nonconvex decentralized gradient descent." IEEE Transactions on signal processing 66.11 (2018): 2834-2848.

---

### Official Review · Reviewer_3u8w · 2022-10-25

**Confidence:** 4
**Correctness:** 4
**Technical Novelty And Significance:** 2
**Empirical Novelty And Significance:** 3
**Recommendation:** 3

**Clarity, Quality, Novelty And Reproducibility:**

Clarity and quality:

Overall the paper is well written. However, the paragraph after Eq 8 could be clarified.

Besides, it is not clear that if lambda = 0 then there is more personalisation. Indeed, the objective becomes constant in this case.

The presentation of personalisation vs federation is a bit confusing and is not discusses theoretically. The main theorem deals with the federated iterate. More generally, one would need more justifications for the approach.

Novelty:

The paper provides one way to deal with composite functionals in a Federated setting, a problem which seems to be new. Otherwise, the proof technique seems rather standard given the assumptions. I mean that, once the model F = sum F_i is formed, the reminder seems to follow from existing analyses in 1. the FBE 2. the FL algorithm used to minimize F. However, the numerical experiments show that at least in practice the approach performs well.

MINOR

"Bregman distrance"

 "unbaised"

"Caculate"

" standrad"

**Strength And Weaknesses:**

Weaknesses

- The gradient of the FBE is actually not a FB step, because it requires the Hessian of the smooth function. So the methods based on the FBE require the Hessian.

-I think that the assumptions A2 and A3 are too strong. They basically say that the first second and third derivatives are bounded. Usually, the smooth term is gradient Lipschitz and we do not assume Lipschitzness of the nonsmooth term. The third derivatives are needed because the Hessian appears in the algo.

- The approach based on the FBE is not justified. Is it a good model to minimize F instead of the true loss f? Why won't we apply a federated version of the Forward Backward algorithm for example? (and then take one additional FB step for personalisation?)


Strength

- Numerical exp show the superiority of the approach over concurrent methods in several contexts.

- The F_i is smooth, so the model can be solved by any FL algorithm which gives a lot of flexibility.

**Summary Of The Paper:**

This paper proposes federated composite optimization, i.e., the nonconvex minimization of sum_i f_i + h in a federated setting. Here, f_i is smooth, h is not smooth but proximable. Note that the f_i are not assumed to be equal to each other (heterogeneous data).

The Forward Backward Envelope (FBE) of f_i + h is a function F_i s.t. one gradient step on F_i is roughly equivalent to one Forward Backward step for f_i + h (it involves the Hessian though). Similarly to the Moreau envelope, the FBE is smooth and depends on a parameter lambda. The paper proposes to minimize F = sum F_i. Tuning lambda allows for more federation or more personalization.

To minimize F, one can use basically any Federated Learning algorithm since the F_i are smooth.
The main theorem upper bounds the squared norm of the gradient of F evaluated at the federated iterate, along the algorithm.

**Summary Of The Review:**

This paper proposes a meta algorithm to solve composite problems in a federated setting. This meta algorithm relies on minimizing the sum of the FBEs with any FL algorithm, because the FBEs are smooth. Once the approximated problem with FBEs is formed, the paper forgets about the original problem (except in the numerical experiment section).

---

> ### Author Response · Authors · 2022-11-19
> **Response to Reviewer 3u8w (1/2)**
>
> Thank you for the comments. We have revised our paper accordingly. All the updates are in blue color.  We address the questions as follows.
>
> 1. Q: "The gradient of the FBE is actually not a FB step, because it requires the Hessian of the smooth function. So the methods based on the FBE require the Hessian."
>
>     A: The gradient of FBE involves an additional Hessian-vector product than the proximal gradient. But the additional computation is not much because it could be calculated through numerical differences between two gradients, see [1]. Specifically, the numerical approximation is based on $\nabla^2 f(x) u \approx (\nabla f(x+tu) - \nabla f(x)) /t$ with small $t > 0$. We note that the proximal gradient step (i.e., ignore the Hessian part) can also serve as an inexact gradient to reduce the computations. Similar ideas are conducted in the personalized FedAvg paper [1].
>
>  2. Q: "I think that the assumptions A2 and A3 are too strong. They basically say that the first second and third derivatives are bounded. Usually, the smooth term is gradient Lipschitz and we do not assume Lipschitzness of the nonsmooth term. The third derivatives are needed because the Hessian appears in the algo. "
>
>     A: The boundedness of the second-order derivatives can be guaranteed if the function itself is of gradient Lipschitz.  The Lipschitz continuity of the Hessian does not mean the function itself is third-order differentiable. It is not a strong condition and covers many applications, e.g., Lasso, matrix completions, and logistic regressions. Similar assumption are used for the FBE based method for composite optimization [2, 3].  The Lipschitz continuity assumption of the nonsmooth term, i.e., the boundedness of subgradients, is also standard in the field of decentralized optimization [4], which is used to bound the consensus error. The $\ell_1$ norm we use in the experiments is with bounded subgradients.
>
>
> 3. Q: "Why won't we apply a federated version of the Forward Backward algorithm for example? (and then take one additional FB step for personalisation?)"
>
>     A: This kind of approach may not have the comparable personalization results to our approach. The personalized FedAvg  paper [1] did similar comparisons between their method and FedAvg with extra gradient step for personalization. We note that our method have the potential to solve FCO with different nonsmooth regularizers and establish similar theoretical guarantees, which the existing methods may fail.
>
> 4. Q: "Is it a good model to minimize F instead of the true loss f?"
>
>    A: The benefits of the FBE comes from the following two aspects. Firstly, the original problem is nonsmooth, the FBE $F$ is smooth although the gradient will involve the Hessian-vector multiplication, while $f$ is nonsmooth. In the centralized setting, under certain mild conditions, both $F$ and $f$ have the same minimum and optimizing with $F$ will leads to many efficient algorithms, see [2,3]. Secondly, the parameter $\lambda$ brings much flexibility. It can not only control the distance between $F$ and $f$, but also control the trade-off between federation and personalization.
>
> 		[1] Fallah, Alireza, Aryan Mokhtari, and Asuman Ozdaglar. "Personalized federated learning: A meta-learning approach." arXiv preprint arXiv:2002.07948 (2020).
>
> 		[2] Stella, Lorenzo, Andreas Themelis, and Panagiotis Patrinos. "Forward–backward quasi-Newton methods for nonsmooth optimization problems." Computational Optimization and Applications 67.3 (2017): 443-487.
>
> 		[3] Themelis, Andreas, Lorenzo Stella, and Panagiotis Patrinos. "Forward-backward envelope for the sum of two nonconvex functions: Further properties and nonmonotone linesearch algorithms." SIAM Journal on Optimization 28.3 (2018): 2274-2303.
>
> 		[4] Zeng, Jinshan, and Wotao Yin. "On nonconvex decentralized gradient descent." IEEE Transactions on signal processing 66.11 (2018): 2834-2848.

---

> > ### Author Response · Authors · 2022-11-19
> > **Response to Reviewer 3u8w (2/2)**
> >
> > We address the other comments from the reviewer as follows:
> >
> > 5. Q: "However, the paragraph after Eq 8 could be clarified. Besides, it is not clear that if $\lambda = 0$ then there is more personalisation. Indeed, the objective becomes constant in this case. "
> >
> >     A: Sorry for the confusion. If $\lambda=0$, then $\theta_i(w) = \arg\min_{\theta_i} \; \nabla f_i(w)^\top (\theta_i - w) + h(\theta_i),$ which is not constant function if $\nabla f_i(w)$ depends on $w$. In the extreme case of linear $f_i$, $\theta_i(w)$ will be a constant function taking value $\arg\min_{\theta_i} f_i(\theta_i) + h(\theta_i)$. Then, $\theta_i(w)$ will be the best personalization parameter and no federation induced. As $\lambda = \infty$ results in only federation, we claim $\lambda \in (0, \infty)$ will allow both federations and personalizations. Other than the linear case, if $f_i$ is a quadratic function with Hessian matrix $\lambda_0 I (\lambda_0 > 0)$, then setting $\lambda = \lambda_0$ will result in the exact personalization and no federation. In this case, a $\lambda \in (\lambda_0, \infty)$ will guarantee both federations and personalizations. We also note that $\lambda > 0$ is needed for the smoothness. Otherwise, the optimization problem in the definition of FBE is not strongly convex and $F_i$ will not be smooth.
> >
> > 6. Q: "The presentation of personalisation vs federation is a bit confusing and is not discusses theoretically. More generally, one would need more justifications for the approach."
> >
> >     A: As explained in the above concern, a proper $\lambda$ will lead to both personalizations and federation under certain assumptions. As in the personalization with the Moreau envelope [5], we can only quantize the amount of the personalization and the federation in the extreme choice of $\lambda$, i.e., $\lambda =0$ and $\lambda = \infty$. Our convergence theorem shows that the solutions $\bar{w}$ (federated parameter) and $\theta_i(w_i)$ (personalized parameter) of pFedFBE are obtainable by the propoblacksed algorithm with the presented complexity.
> >
> > 7. Q: "MINOR typos."
> >
> >     A: We have modified the typos.
> >
> > [5] T Dinh, Canh, Nguyen Tran, and Josh Nguyen. "Personalized federated learning with moreau envelopes." Advances in Neural Information Processing Systems 33 (2020): 21394-21405.

---

> > > ### Comment · Reviewer_3u8w · 2022-12-05
> > > **Thanks for the answers**
> > >
> > > Thanks for the answers.
> > >
> > > This paper proposes a new objective for FL based on FBE.
> > >
> > > I am satisfied with the answers of Q3 and Q4. However, FL with Moreau envelopes was considered previously and the approach with FBE here is not as clean as the approach with Moreau envelopes: the resulting algorithm is actually a second order algorithm (without the benefits of second order methods such as quasi Newton), which also implies to make assumptions on the Hessian.
> > >
> > > I know that the fundamental reason is that FBE does not represent the proximal gradient algo as good as Moreau envelope represents the proximal point algo. But I don't see enough impact for this combination of FL + FBE to be published in ICLR

---

### Official Review · Reviewer_bfG5 · 2022-10-26

**Confidence:** 3
**Correctness:** 2
**Technical Novelty And Significance:** 2
**Empirical Novelty And Significance:** 2
**Recommendation:** 5

**Clarity, Quality, Novelty And Reproducibility:**


The paper is written clearly and easy to follow. The authors can highlight their contributions better by stressing the differences between FedFBE and pFedMe in the algorithm derivation and convergence proof.

**Strength And Weaknesses:**

Strength:
the topic is important for distributed learning and federated learning. This paper proposed an interesting method to obtain explicit forms of gradients. A convergence proof is performed to justify the algorithm's efficacy. The paper is not hard to follow.
Weaknesses:
My major concern is the novelty. It seems that the proposed method uses the framework of pFedMe but estimates the local gradient using an explicit form, plus additional computation on the Hessian matrix. Although the experimental result of FedFBE shows faster convergence over other methods, it is not fully convincing to say that FedFBE has higher efficiency. Estimating the Hessian matrix definitely helps with convergence, but it also leads to the consumption of much more computation. pFedMe doesn't use the explicit form partly because it can save computational power by avoiding that Hessian matrix. Therefore, I am not sure that the problem will be solved without creating another bigger problem. Thus, the novelty in terms of the solution technique is thus limited.
I wonder about the influence of hessian components of the proposed method on the final results. Some ablation studies can be done to justify the efficacy of FBE.

**Summary Of The Paper:**

In this paper, the authors propose a novel personalized method for FCO via the forward-backward envelope. Personalized models are firstly updated, then the local models are updated with Second-order optimization. After the local update, the global update is done via FedAVG. Convergence results of the proposed algorithm are shown under similar assumptions in pFedMe. Numerical experiments on the federated lasso, federated matrix completion, and nonsmooth deep neural network were done.

**Summary Of The Review:**

Generally speaking, the topic of the paper is of importance and interest. However, the authors can improve the work by addressing the above-mentioned concerns, especially about the computational efficiency and the ablation studies in the second-order info in the optimization.

---

> ### Author Response · Authors · 2022-11-19
> **Response to Reviewer bfG5**
>
> Thank you for the comments. We have revised our paper accordingly. All the updates are in blue color.  We address the questions as follows.
>
> 1. Q: “Estimating the Hessian matrix definitely helps with convergence, but it also leads to the consumption of much more computation. pFedMe doesn't use the explicit form partly because it can save computational power by avoiding that Hessian matrix. Therefore, I am not sure that the problem will be solved without creating another bigger problem. Thus, the novelty in terms of the solution technique is thus limited.”
>
>     A: The gradient of FBE involves an additional Hessian-vector product than the proximal gradient. But the additional computation is not much because it could be calculated through numerical differences between two gradients, see [1]. Specifically, the numerical approximation is based on $\nabla^2 f(x) u \approx (\nabla f(x+tu) - \nabla f(x)) /t$ with small $t > 0$. We note that the proximal gradient step (i.e., ignore the Hessian part) can also serve as an inexact gradient to reduce the computations. Similar ideas are conducted in the personalized FedAvg paper [1].
>
> [1] Fallah, Alireza, Aryan Mokhtari, and Asuman Ozdaglar. "Personalized federated learning: A meta-learning approach." arXiv preprint arXiv:2002.07948 (2020).
>
> 2. Q: “I wonder about the influence of hessian components of the proposed method on the final results. Some ablation studies can be done to justify the efficacy of FBE.”
>
>     A: The personalized FedAvg paper [1] did similar comparisons on their methods with and without Hessian information, it shows the use of Hessian information leads to better test accuracies. The situation of our case is similar and we do not put these results due to the page limit.
>
> 3. Q: “The authors can highlight their contributions better by stressing the differences between FedFBE and pFedMe in the algorithm derivation and convergence proof.”
>
>     A: Thanks for the suggestions. We have highlighted the differences behind Eq. (6) and in the paragraph behind Assumption (A5).

---

### Decision · Program_Chairs · 2023-01-20

**Decision:**

Reject

**Justification For Why Not Higher Score:**

The writing of the paper should be improved significantly based on the reviewers' remarks. The scalability concerns should be addressed as well.

**Justification For Why Not Lower Score:**

N/A

**Metareview: Summary, Strengths And Weaknesses:**

The work proposes a personalization approach for federated learning (FL) using the forward backward envelope (FBE) of the clients' loss functions.

While the approach is interesting, its superiority beyond the existing approaches (such as with Moreau envelopes) is unclear. Moreover, it looks like the overall approach is a second order approach, which has scalability and storage issues without the presence of quasi-Newton type approximations.